# Smooth muscle cell Piezo1 depletion results in impaired contractile properties in murine small bowel
Geoanna M. Bautista [1,2,8], Yingjie Du[3,8], Michael J. Matthews[4], Allison M. Flores[4], Nicole R. Kushnir [4], Nicolle K. Sweeney[4], Nam Phuong N. Nguyen [4], Elmira Tokhtaeva[4], R. S. Solorzano-Vargas[4], Michael Lewis[5], Matthias Stelzner[6], Ximin He [3], James C. Y. Dunn [7] & Martin G. Martin [4] ✉

Piezo1 is a mechanosensitive cation channel expressed in intestinal muscularis cells (IMCs), including smooth muscle cells (SMCs), interstitial cells of Cajal, and Pdgfrα+ cells, which form the SIP syncytium, crucial for GI contractility. Here, we investigate the effects of SMC-specific Piezo1 deletion on small bowel function. Piezo1 depletion results in weight loss, delayed GI transit, muscularis thinning, and decreased SMCs. Ex vivo analyses demonstrated impaired contractile strength and tone, while in vitro studies using IMC co-cultures show dysrhythmic $Ca^{2+}$ flux with decreased frequency. Imaging reveal that Piezo1 localizes intracellularly, thereby likely impacting $Ca^{2+}$ signaling mechanisms modulated by $Ca^{2+}$-handling channels located on the sarcoplasmic reticulum and plasma membrane. Our findings suggest that Piezo1 in small bowel SMCs contributes to contractility by maintaining intracellular $Ca^{2+}$ activity and subsequent signaling within the SIP syncytium. These findings provide new insights into the complex role of Piezo1 in small bowel SMCs and its implications for GI motility.

The gastrointestinal (GI) tract layers exhibit distinct coordinated mechanical properties that enable efficient propulsion to facilitate nutrient assimilation[1–5]. This complex process involves various cell types, including smooth muscle (SMC), interstitial cells of Cajal (ICC), and Pdgfrα+ cells, which form the SIP syncytium[6–8] which helps integrate signals from enteric neurons (EN) and modulates GI motility. SMCs, the primary effectors of GI motility, respond to direct mechanosensory stimulation and signaling via electrical coupling with the surrounding interstitial cells. The cells within the SIP syncytium depend on the dynamic $Ca^{2+}$ signaling to regulate SMC excitability as part of the myogenic, or, more recently termed, "SIPgenic," modulation of GI motility[6]. However, the molecular mechanisms that allow the SIP syncytium to sense mechanical forces remain poorly defined[9,10].

Piezo1, a mechanosensitive nonselective cation channel, has been implicated in various aspects of intestinal epithelial function[11–14]. The selective deletion of Piezo1 in the GI epithelium results in a decline in 5-HT-containing cells and is associated with prolonged gut transit time[14]. Epithelial Piezo1 is known to regulate goblet cell mucus production, impacting intestinal transit[13]. Yet, despite the presence of Piezo1 in all of the SIP cells and ENs that comprise the intestinal muscularis cells (IMCs), there is limited understanding regarding its function in the mechanosensation of GI motility[15–17].

The multilevel integration necessary to coordinate SMCs and generate contractile, rhythmic forces relies heavily on highly regulated $Ca^{2+}$ signaling that responds to the mechanically triggered functions of the intestine[6–8]. Notably, Piezo1 plays a crucial role in modulating $Ca^{2+}$ dynamics and activating downstream signaling pathways in various mechanically active organs[18,19]. Given its ubiquitous expression in cell types governing motility, including SMC, ICC, and Pdgfrα+ cells, Piezo1 will likely contribute to the multicellular regulation of GI motility. This may occur through potential cell-to-cell interactions within the SIP syncytium, specifically enabling

[1]Department of Pediatrics, Division of Neonatology, University of California Davis Children's Hospital, Sacramento, CA, 95817, USA. [2]Department of Pediatrics, Division of Neonatal-Perinatal Medicine, Mattel Children's Hospital and the David Geffen School of Medicine, University of California Los Angeles, Los Angeles, CA, 90095, USA. [3]Department of Materials Science and Engineering, University of California Los Angeles, Los Angeles, CA, 90095, USA. [4]Department of Pediatrics, Division of Gastroenterology and Nutrition, Mattel Children's Hospital and the David Geffen School of Medicine, University of California Los Angeles, Eli and Edythe Broad Center of Regeneration Medicine and Stem Cell Research, Los Angeles, CA, 90095, USA. [5]Department of Pathology, VA Greater Los Angeles Healthcare System, Los Angeles, CA, 90073, USA. [6]Department of Surgery, VA Greater Los Angeles Healthcare System, Los Angeles, CA, 90073, USA. [7]Division of Pediatric Surgery, Departments of Surgery and Bioengineering, Stanford University School of Medicine, Stanford, CA, 94305, USA. [8]These authors contributed equally: Geoanna M. Bautista, Yingjie Du. ✉e-mail: mmartin@mednet.ucla.edu

SMCs of the intestine to respond to its mechanical needs. This is especially true given Piezo1's mechanosensitive properties, fast activation, slow-intermediate inactivation, and nonselective cation permeability, which are all crucial to the intrinsic functions of the gut[7,20].

Advancements in the in vitro culturing of IMCs, including the SIP cells, allow us to study the complex interactions driving gut motility[21–23]. In this setting, IMCs generate spontaneous cyclic contractions for extended lengths of time using a unique co-culture system developed in our lab[21]. Furthermore, tunable stimuli-responsive hydrogels provide a unique opportunity to investigate the effects of stretching on IMCs. Specifically, temperature-stimuli-responsive scaffolds may induce localized mechanical tissue stimulation in an in vitro setting[19,24,25]. Moreover, ex vivo assessments of isolated full-thickness bowel segments using multi-wire myography can provide insight into the myogenic (or "SIPgenic"[6]) stimuli that lead to intestinal contractility without extrinsic neurogenic input[26,27]. These tools and models have made it feasible to explore the consequences of Piezo1-deficient SMCs on the mechanosensation of intestinal contractility.

Our study reveals that targeted deletion of Piezo1 in SMCs results in impaired murine growth, significant attenuation of homeostatic motility, thinning of the external muscularis with altered expression of cell types within the SIP syncytium, and faulty intracellular $Ca^{2+}$ signaling. These findings highlight the critical role of Piezo1 in regulating intestinal motility and underscore the importance of further research into the mechanisms underlying Piezo1-mediated mechanosensation.

## Results

### Inducible deletion of Piezo1 in enteric SMCs

To isolate the role of PIezo1 expressed in enteric SMCs, we generated homozygote knockout (KO), Piezo1$^{fl/fl}$ - Myh11$^{ERT2/Cre}$ (Piezo1$^{\Delta SMC}$) mice, which were compared to wildtype (WT), Piezo1$^{WT/WT}$ - Myh11$^{ERT2/Cre}$ (Piezo1$^{WT}$)[28]. To specify cells with Cre-induced nuclear translocation, mice were also generated with mTmG reporter expressing membrane-targeted tdTomato+ (mT) or EGFP (mG) (Piezo1$^{WT;mTmG}$ and Piezo1$^{\Delta SMC;mTmG}$)[29]. Tamoxifen (Tam, 50 mg/kg × 5 d) was given via orogastric gavage (OG) to 4–6-week-old mice. Unless otherwise specified, distal small bowel samples were obtained and analyzed between 21 and 28 days after the last dose of Tam. In the absence of Tam, the various intestinal layers of Piezo1$^{WT;mTmG}$ mice were mT+ in transverse-sections of whole mount (WM) preparations (Fig. 1A). To confirm the specificity of the Cre-induced nuclear translocation, Piezo1$^{WT;mTmG}$ mice had restricted mG+ cells in the subepithelial myofibroblast and SMCs of muscularis[29] (Fig. 1B; Supplementary movie).

The external muscularis layer was dissected from the whole intestine to determine appropriate layer-specific Cre-induced DNA cleavage[30]. Intestinal muscularis tissue from Piezo1$^{\Delta SMC}$ mice showed the expected reduction in DNA, mRNA, and protein expression as determined by PCR, RT-qPCR, and Western blot compared to Piezo1$^{WT}$ (Supp. Fig. 1A-B) [OBJ][OBJ]. Piezo1 mRNA expression was similarly reduced in the muscularis layers of the stomach, large intestine, and full-thickness bladder in Piezo1$^{\Delta SMC}$; however, no obvious gross abnormalities were observed (Fig. 1C). Immunofluorescent (IF) imaging with an anti-Piezo1 and β-actin antibodies on frozen transverse-section preparations showed an abundance of Piezo1 in Piezo1$^{WT}$ muscularis and significant depletion in Piezo1$^{\Delta SMC}$ mice (Fig. 1D [OBJ][OBJ].

We then assessed whether the loss of Piezo1 in SMCs impacted Piezo2. Piezo2 mRNA was slightly increased in the bladder ($p < 0.0449$) and more significantly in the muscularis layers of the small intestine in Piezo1$^{\Delta SMC}$ mice ($p < 0.0001$) (Fig. 1C). Piezo2 staining of WM samples was undetectable in the SMCs of Piezo1$^{WT;mTmG}$ and Piezo1$^{\Delta SMC;mTmG}$ mice and limited to non-SMC cell types (Supp. Fig. 1C).

### SMC-specific depletion of Piezo1 results in weight loss and impaired whole-gut transit

We assessed body weight and intestinal transit following Tam. We found that Piezo1$^{\Delta SMC}$ mice show weight loss starting 14 days post-Tam induction (Fig. 1E). We measured whole gut transit time using the carmine red dye methylcellulose challenge, recording the time to the first discolored stool pellet[31]. Compared to control Piezo1$^{WT}$ mice, Piezo1$^{\Delta SMC}$ mice had significantly prolonged transit time, with more than double the duration (Fig. 1F). Piezo1$^{\Delta SMC}$ mice had fewer stool pellets than Piezo1$^{WT}$ mice during 8-h and reduced chow consumption compared to Piezo1$^{WT}$ mice (Fig. 1G, H). These changes occurred without gross differences in small bowel diameter or length between Piezo1$^{WT}$ and Piezo1$^{\Delta SMC}$ mice (Supp. Fig. 1D-E). These findings suggest that Piezo1 expression in SMCs is crucial in maintaining appropriate weight gain and normal gut luminal transit.

### Loss of SMC Piezo1 leads to impaired contractility in small bowel ex vivo

To investigate how Piezo1-deficient SMCs impacted intestinal contractility, we conducted ex vivo isometric force assessments on intact, full-thickness bowel ring segments using multi-wire myography[27] (Fig. 2A). During steady state (SS), Piezo1$^{\Delta SMC}$ mice demonstrated reduced amplitude (Fig. 2B) and area under the curve (AUC) (Fig. 2C), shorter contractile duration (Fig. 2D), and longer, more irregular periods between contractions compared to Piezo1$^{WT}$ controls (Coefficient of variation, CoV 29% vs. 6.2%) (Fig. 2E). This data suggests that loss of Piezo1 in SMCs alters phasic contractions potentially contributing to the delay in transit seen in vivo.

Since Cav1.2 channels primarily contribute to the $Ca^{2+}$ influx necessary to trigger contractions in intestinal SMCs[32,33], we investigated the contribution of Piezo1 in SMCs in the setting of nicardipine, a Cav1.2 inhibitor. Inhibition with nicardipine reduced contraction amplitude in both groups but more profoundly in Piezo1$^{\Delta SMC}$ bowel (Fig. 2F, G). Adding GsMTx4, the Piezo1/2 and Trpc antagonist[34], reduced amplitude from baseline in both groups but less than with nicardipine alone (Fig. 2F, G). Combined nicardipine and GsMTx4 further reduced amplitude in Piezo1$^{WT}$ but not in Piezo1$^{\Delta SMC}$ segments. Notably, nicardipine did not affect contractile patterns in Piezo1$^{WT}$ segments but shortened contraction duration in Piezo1$^{\Delta SMC}$ segments (Fig. 2H, I). This suggests that Piezo1 modulates the strength and pattern of contractions in response to mechanical forces.

### Altered small bowel tissue properties in Piezo1- depleted SMCs

We then hypothesized that loss of Piezo1 in the small bowel would impair length-tension, viscoelastic, and stress-strain properties in response to active stretch (Fig. 2A). Piezo1$^{\Delta SMC}$ mice exhibited lower passive tension (tonicity) and more significant reduction in maximum force at SS compared to control mice (Fig. 3A–D). Piezo1$^{\Delta SMC}$ mice also had a more rapid relaxation rate, particularly in the first 5 to 30 s following acute stretch (Fig. 3G–J). To isolate potential contributions of the enteric nervous system (ENS), tetrodotoxin (TTX, a neuronal sodium channel inhibitor that prevents action potentials in neurons) (Fig. 3B–E) and L-NNA (an nNOS inhibitor) combined with ODQ (soluble guanylyl cyclase (sGC) inhibitor)[6] were applied (Fig. 3C–F). While the addition of TTX or L-NNA/ODQ did not impact the directionality in the length-tension relationship in Piezo1$^{\Delta SMC}$ mice (Fig. 3B, C), they reduced the difference in force reduction, suggesting a potentially complex relationship between the ENS and Piezo1 function in SMCs (Fig. 3E, F).

### SMC-specific depletion of Piezo1 impairs $Ca^{2+}$ flux in co-cultured intestinal muscularis cells (IMCs)

To investigate the impact of Piezo1-deficient SMCs on the surrounding syncytium responsible for contractility, we employed our validated method for culturing IMCs while preserving the various cell types (SMCs, ICCs, Pdgfrα + , ENs) in vitro[19]. IMCs from Piezo1$^{WT}$ and Piezo1$^{\Delta SMC}$ pups were treated with 4-hydroxytamoxifen (4-OHT, 0.1 μM) and transduced with gCAMP6f and mCherry lentiviruses to assess $Ca^{2+}$ flux and displacement of IMCs[19,35]. Appropriate Piezo1 gene cleavage and reduced mRNA levels were confirmed in cells from the Piezo1$^{\Delta SMC}$ compared to control Piezo1$^{WT}$ (Supp. Fig. 2A-B)[33].

To determine whether stretch-induced $Ca^{2+}$ changes occur in an SMC-Piezo1-dependent manner in vitro, a temperature-sensitive (TS) stretch-inducible hydrogel scaffold that mimicked the elastic modulus of native

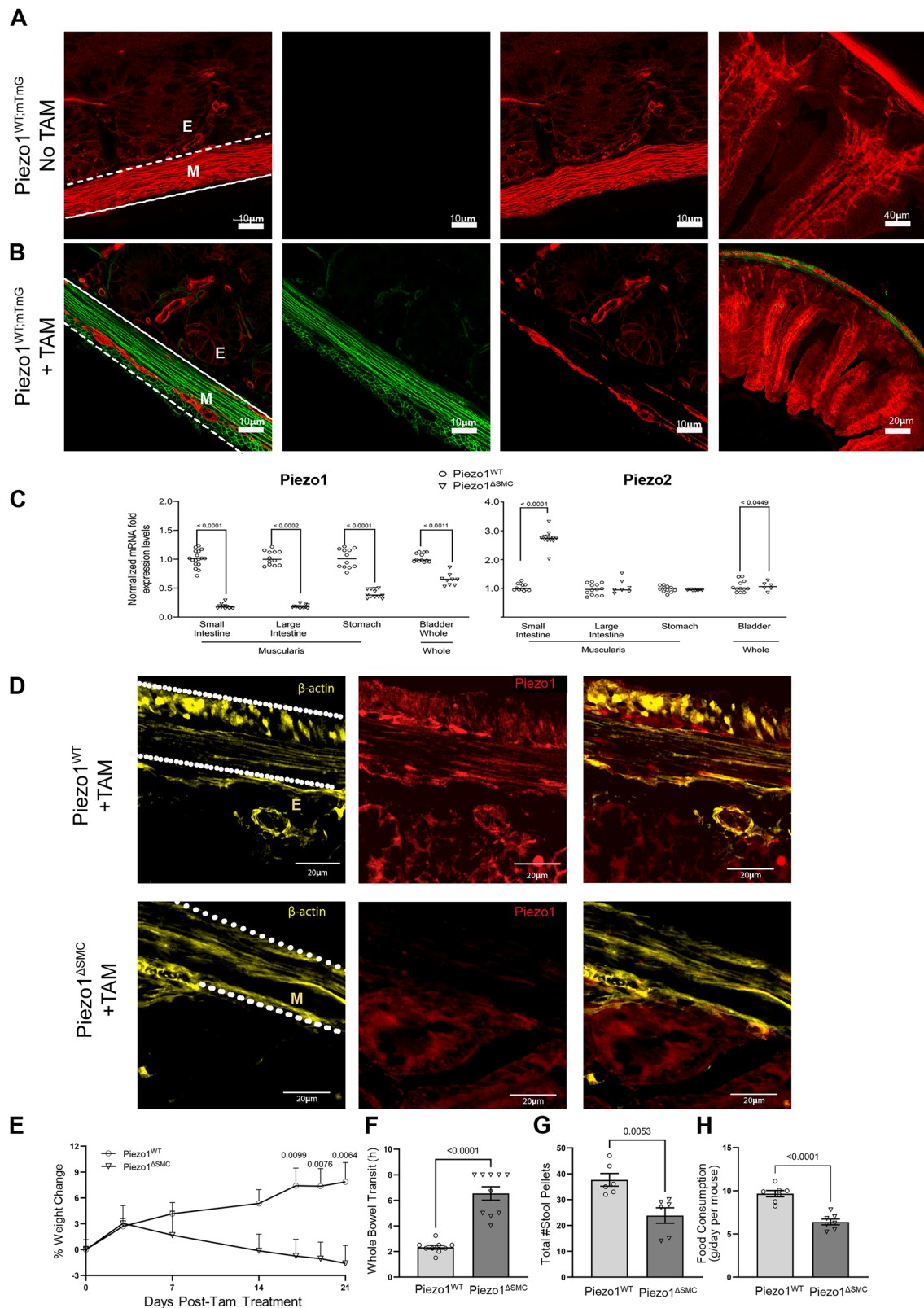

bowel tissue (4.1kP) was developed (Fig. 4A–C). Correlation between contractile activity and Ca²⁺ flux was confirmed by comparing cell movement compared to Ca²⁺ flux (Fig. 4D, E). Piezo1$^{WT}$ IMCs exhibited rhythmic, spontaneous contractions, which doubled in frequency when stretched despite the decrease in temperature. Piezo1$^{\Delta SMC}$ IMCs instead displayed irregular Ca²⁺ flux that was significantly decreased, declining further with

stretch (Fig. 4F, G). Furthermore, the global knockdown of Piezo1 generated by transduction with Piezo1shRNA lentivirus resulted in a complete cessation of Ca²⁺ flux (Fig. 4H, I).

We then investigated the effects of Piezo1 agonist (Yoda1) and antagonist (GsMTx4) on Ca²⁺ flux in IMCs[36,37] (Fig. 5A, B). Yoda1 increased the frequency of Ca²⁺ flux in Piezo1$^{WT}$ IMCs but failed to restore the Ca²⁺

**Fig. 1 | Inducible SMC-specific Piezo1 depletion results in poor growth and delayed bowel transit.** Efficient Tam-induced translocation of Cre-ER into the nucleus was confirmed in distal small bowel segments taken from 7-9 week-old mice following Tam administration at 4-6 weeks old, with all assessments performed age-matched in both groups. **A** Myh11-Cre expression in the distal small bowel segment from Piezo1^WT mice with membrane-associated Tdtomato fluorescence (mTmG) reporter (Piezo1^WT;mTmG) shows exclusive Tdtomato fluorescence in membranes of all cell types across the layers of the bowel in transverse-sections of WM preparations, indicating an absence of Cre expression or leakage without Tam induction. **B** In contrast, similarly aged mice post-Tam treated Piezo1^WT;mTmG mice, the epithelial layer retained Tdtomato, while the cellular components of the muscularis were either Tdtomato or GFP +. Leica Confocal SP8-STED microscope with scale bars set at 10, 20, and 40 μm. **C** mRNA expression of Piezo1 and Piezo2 in Piezo1^ΔSMC and Piezo1^WT mice from muscularis samples isolated from the distal small intestine, proximal colon and stomach, and full-thickness samples from the bladder. qPCR was performed using the ΔΔCT method relative to *Gapdh* expression. Multiple t-test

analyses with Welch's correction were performed with a False Discovery Rate (FDR) set to 5% with the corrected p-values as indicated above comparison groups. Data are shown as mean ± SEM, n = 4–6 mice per group. **D** Frozen transverse-section of distal small bowel from Piezo1^WT (upper) and Piezo1^ΔSMC (lower) mice post-Tam, stained with anti-β-actin (yellow) and Piezo1 (Alamone Labs; red) antibodies and imaged with a Zeiss Confocal LSM880 at 40× magnification with 20 μm scale bars. The parallel dashed lines indicate the location of the muscularis (M). Beneath the bottom line is the epithelial (**E**) layer. **E** Body weight at various days post-Tam treatment in Piezo1^WT and Piezo1^ΔSMC mice. **F** Carmine-red dye was given via oral gavage, and the time to the first red stool pellet was recorded to indicate whole bowel transit time in both groups of mice at 7–9 weeks old following tamoxifen administration at 4-6 weeks old. Total number of hours capped at 8 h. **G** A total number of stool pellets was produced after 8 h in both groups of mice. **H** Daily food consumption in both groups of mice. Two-tailed t-test with Welch's correction performed with a significance set at *p* < 0.05 is shown above. Data are shown as mean ± SEM, N = 6–10 mice per group.

flux in Piezo1^ΔSMC. GsMTx4 halted the Ca^{2+} flux in both Piezo1^WT and Piezo1^ΔSMC. Adding carbachol and 5-HT increased Ca^{2+} flux in Piezo1^WT and Piezo1^ΔSMC (Fig. 5A, B). Furthermore, extracellular Ca^{2+} was required for Ca^{2+} flux in both Piezo1^WT and Piezo1^ΔSMC IMCs at baseline and Yoda1-induced Ca^{2+} influx in both groups. These findings suggest that while Piezo1 in SMCs is essential for stretch-induced stimulation, it is not required for cholinergic and serotonergic activation of Ca^{2+} flux, confirming an alternative contractility pathway in IMCs. Similar responses to Yoda1 and GsMTx4 were displayed in IMCs extracted from human fetal intestinal samples at 17 weeks gestational age, suggesting potentially overlapping functions of Piezo1 in mice and humans (Supp. Fig. 2C, D).

Overall, these data show that in an isolated in vitro model of IMCs, loss of Piezo1 in SMCs results in a disruption in Ca^{2+} flux within the intact SIP syncytial cells in this model. However, due to the inherent limitations of this approach, it is difficult to determine the global impact on electrical coupling or contractility.

### Loss of Piezo1 in SMCs leads to morphological and cellular changes in the small bowel external muscularis

To assess if histologic changes contributed to the dysmotility in Piezo1^ΔSMC mice, we examined the distal small bowel muscularis. Piezo1^ΔSMC mice had a significantly thinner muscularis at 21 d post-Tam in formalin-fixed paraffin-embedded (FFPE) samples (Fig. 6A). We used WM preparations from mTmG reporter mice to then confirm changes in the SMC population in the muscularis following Tam administration. The density of SMCs was significantly reduced when compared to the circular and longitudinal layers of the Piezo1^WT;mTmG muscularis (Fig. 6B; Supp. Fig. 3). Despite thinning of the muscularis and reduced SMC density in Piezo1^ΔSMC mice, expression of the apoptotic marker, activated cleaved caspase3 (CC3) was similar to that in the muscularis of Piezo1^WT mice (**Supp. data file**).

Next, we examined the abundance of the other cellular components by imaging WM samples. Staining of ICC, Pdgfrα +, and glial cells using anti-c-Kit, Pdgfrα, and Gfap antibodies showed a slight yet significant increase in the density of these cell populations in Piezo1^ΔSMC compared to Piezo1^WT muscularis (Fig. 6C–E; Supp. Fig. 4). Similarly, staining of the neuronal bundles, fibers, and cells using an anti-Tubb3 antibody was slightly higher in Piezo1^ΔSMC than in Piezo1^WT muscularis (Fig. 6F–H; Supp. Fig. 4). These data highlight the loss of Piezo1 in SMCs results in a significant thinning of the muscularis and an associated reduction in the cellular density of SMCs and a proportional increase in the non-SMC cellular populations in the muscularis layers.

### Piezo1 uniquely localizes intracellularly in SMCs of the intestinal muscularis

Next, we determined Piezo1 cellular localization by obtaining high-resolution confocal imaging of WM preparation obtained from Piezo1^WT;mTmG or Piezo1^ΔSMC;mTmG mice stained with an anti-Piezo1 antibody. Piezo1 was abundant in the SMCs of Piezo1^WT;mTmG mice and selectively

depleted in the SMCs in the muscularis layer of Piezo1^KO;mTmG mice (Fig. 7A). High magnification confirms punctate Piezo1+ clusters located primarily within membranous organelles rather than the plasma membrane of SMCs (Fig. 7B, C). These data suggest that Piezo1 in the SMCs of the external muscularis largely localizes intracellularly and may have a more complex role in these cells than previously understood.

### Piezo1-deficient SMCs alter the expression of known modulators of intestinal motility

To further characterize the changes driven by the loss of Piezo1 in SMCs, we examined the channels essential for Ca^{2+} homeostasis and motility by assessing their relative mRNA using qPCR and protein localization using IF staining of WM preparations. Transcript levels and protein localization of the store-operated Ca^{2+} channels (Orai1 and Orai3) and the TRP channel family member, Trpc4 were decreased in Piezo1^ΔSMC compared to Piezo1^WT (Fig. 8A, B; Supp. Figure 5A–C; Supp. Fig. 6A). There was an increase in mRNA levels of Cav1.2, the L-type channel critical for Ca^{2+} influx in intestinal SMCs, confirmed by IF showing a greater abundance in the muscularis layers of Piezo1^ΔSMC mice (Fig. 8A, B; Supp. Fig. 5A, Supp. Fig. 6B). Furthermore, the Ca^{2+}-activated Cl^- channel Ano1 typically expressed in ICCs increased 10-fold in Piezo1^ΔSMC compared to that in Piezo1^WT muscularis (Fig. 8A, B; Supp. Fig. 5A-B). To confirm that this increase was specific to SMCs, we obtained high-resolution imaging of WM stained with anti-Ano1 and c-Kit (ICC) antibodies and found a significant increase in Ano1 in SMCs of Piezo1^ΔSMC mice (Fig. 8C). Collectively, these data suggest that the loss of Piezo1 in SMCs impacts the levels and cell-type localization of other Ca^{2+} related channels located within the cells of the SIP syncytium that contribute to the maintenance of gut motility.

### Piezo1 in SMCs colocalizes with proteins anchored to the sarcoplasmic reticulum

To better define the intracellular role of Piezo1 and its potential localization to the sarcoplasmic reticulum (SR), we performed confocal colocalization analyses using antibodies against ryanodine receptor (RyR) and inositol 1,4,5-trisphosphate receptor (IP3R) in SMCs of external muscularis. Piezo1 is colocalized with RyR and IP3R, indicating its presence in the SR, a crucial intracellular Ca^{2+} store that contributes to excitation-contraction coupling (Fig. 9A, B; Supp. Fig. 6A-B). Interestingly, while RyR abundance remained similar in both Piezo1^WT;mTmG and Piezo1^ΔSMC;mTmG mice, there was a significant reduction in IP3R expression in Piezo1^ΔSMC;mTmG compared to that in Piezo1^WT;mTmG SMCs (Fig. 9A, B). This differential impact on Ca^{2+}-release channels suggests that Piezo1 may have a role in maintaining IP3R levels during homeostasis[38].

### Loss of SMC Piezo1 impairs responsiveness to carbachol, an IP3R channel-dependent agonist

Using isometric force assessment, we found that carbachol rapidly increased active tension in Piezo1^WT mice without affecting contraction duration or

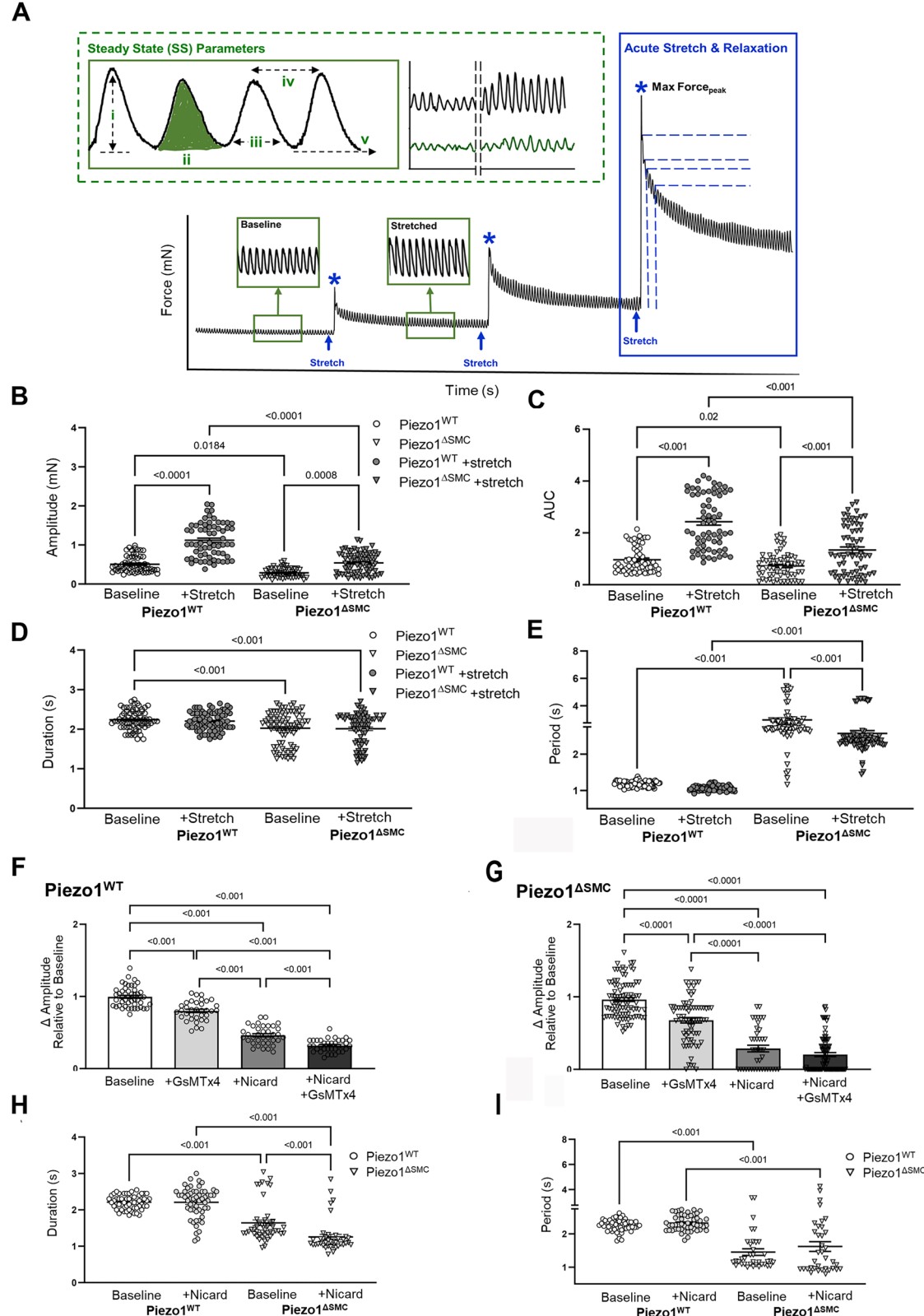

period (Fig. 9C–E). However, this carbachol-induced tension was significantly attenuated in Piezo1$^{\Delta SMC}$ mice, consistent with the observed decline in IP3R expression following Piezo1 depletion (Fig. 9B). Interestingly, in IMCs isolated from murine pups, carbachol restored Ca$^{2+}$ signaling in vitro (Fig. 5A, B), a possible Piezo1-independent, age-dependent pathway.

## Discussion

The loss of Piezo1 in SMCs of the small bowel leads to significant alterations in intestinal motility and function (**graphical abstract**), as evidenced by (1) delayed gut transit and weight loss without an increase in mortality; (2) thinning of the external muscularis and a decline in SMC density; (3) diminished contractile pattern and strength; (4) alterations in tissue

**Fig. 2 | Impaired contractile properties with loss of Piezo1 in SMCs.** Isotonic force measurements were obtained from distal ileal ring segments (3 mm long) from 4- to 6-week-old mice at least 21 days post-Tam at baseline and following acute stretch using a multi-wire myograph. **A** General schematic providing an overview of the experimental approach. Green dotted box illustrating assessments performed at SS (green) or once the contractions have plateaued following stretch: **i.** amplitude (mN), the active or contractile force (height of contraction from passive force (v)), **ii.** AUC (mN-s) measuring the work, **iii.** duration (s), or time of the entire contraction, **iv.** period (s) or time between peaks was used to quantify dysrhythmic patterns observed during trials. Representative tracings of Piezo1$^{WT}$ (black) and Piezo1$^{\Delta SMC}$ (green). Phasic contractile activity, including (**B**) amplitude, (**C**) AUC, (**D**) duration, and (**E**) period, were altered in Piezo1$^{\Delta SMC}$ mice compared to controls ($n$ = 50 samples, $N$ = 3 mice per group). The effects of GsMTx4 and Nicardipine inhibitor exposure on the contraction amplitudes for (**F**) Piezo1$^{WT}$ and (**G**) Piezo1$^{\Delta SMC}$ groups. **H** Duration and (**I**) period changes in contraction behavior when Piezo1$^{WT}$ and Piezo1$^{\Delta SMC}$ are only exposed to nicardipine. For **F–I**, $n \geq 50$ data points from N = 3 mice per group were analyzed with one-way ANOVA with multiple comparison tests.

properties including tonicity and viscoelastic properties; (5) dysregulated expression of ion channels in SMCs (Ano1, Cav1.2, Trpc4, Orai1/3); and (6) predominant localization of Piezo1 in the SR/ER of SMCs with reduced IP3R expression.

Unlike other SMC-specific knockout models that often lead to severe morbidities, such as *Cav1.2, Trpc4/6 and Yap/Taz*[27,32,39,40], Piezo1$^{\Delta SMC}$ mice exhibited milder effects, suggesting a more nuanced role for Piezo1 in intestinal homeostasis. The thinning of the muscularis in Piezo1$^{\Delta SMC}$ mice occurred without significant cellular apoptosis or bowel distension (Fig. 6A; Supp. Fig. 1D, E), likely indicating either remodeling or a decline in the replacement of SMCs [OBJ][OBJ]. This remodeling likely stems from altered intracellular Ca$^{2+}$ dynamics, affecting SMC contractile functions and interactions within the SIP syncytium. The dysregulated Ca$^{2+}$ signaling may then attenuate small bowel tonicity, impairing the maintenance of normal muscularis architecture. These alterations in Ca$^{2+}$ flux could further impact cross-bridge cycling, myosin light chain phosphorylation, and SMC responses to internal cellular traction forces leading to [OBJ][OBJ]. The resulting muscularis thinning would result in fewer contractile proteins per unit length, further weakening the force-generating properties observed in Piezo1$^{\Delta SMC}$ mice (Figs. 2–3).

Interestingly, despite the overall thinning of the muscularis layer, we observed a slight, but significant increase in the density of enteric neurons, ICCs, and Pdgfrα+ cells (Fig. 6; Supp. Figure 3). This retention of specialized SIP syncytium cells suggests that the observed contractile abnormalities may stem from impaired SMC receptiveness, rather than depletion of these crucial cellular components important for contractility. Furthermore, the unique impact of Piezo1-deficient SMCs on contraction patterns distinguishes this model from other Ca$^{2+}$ channels. For instance, SMC-specific depletion of Trpc4 and Trpc6 disrupts excitation-contraction coupling while preserving spontaneous rhythmic contractions[39]. In contrast, the loss of Piezo1 in SMCs appears to have more far-reaching effects on contractile function and muscularis properties.

Given the functional coupling between the SMCs and ICCs and Piezo1 expression in both cell types[6], we hypothesize that loss of Piezo1 in SMCs may impair intracellular Ca$^{2+}$ stores and signaling mechanisms. This disruption could then affect the transmission and propagation of signals from ICCs, explaining the observed alterations in contractile patterns and strength. In support of this hypothesis, we observed significant changes in the expression of channels required for Ca$^{2+}$ homeostasis and contractility following the loss of Piezo1 (Fig. 8; Supp. Figs. 5–6; **graphical abstract**)[41]. Most notably, there was a 10-fold increase in Ano1 expression in the muscularis of Piezo1$^{\Delta SMC}$ compared to control Piezo1$^{WT}$ mice, with unexpected abundance in SMCs[41–47]. These findings, coupled with the increased expression of Cav1.2, the L-type channel critical for Ca$^{2+}$ influx in intestinal SMCs, suggest a potential compensatory mechanism to address impaired intracellular Ca$^{2+}$ activity or perception of adequate Ca$^{2+}$ stores.

The intracellular localization of Piezo1 in small bowel SMCs, particularly its colocalization with channels anchored to the SR/ER (Figs. 7–9, Supp. Fig. 6), supports a role in intracellular Ca$^{2+}$ dynamics. This unique positioning would enable Piezo1 to interact with various Ca$^{2+}$-handling proteins within specialized nanodomains, contributing to the intricate signaling networks controlling SMC contractility and function. While Piezo1 has traditionally been viewed as a plasma membrane-bound channel, emerging evidence suggests it also functions within subcellular organelles across other mechanically active organs[48].

Recent studies have expanded our understanding of Piezo1's cellular distribution, identifying its presence in the ER, nucleus, and other intracellular sites[48]. Furthermore, the localization of these mechanosensors appears to be dynamic, potentially shifting in response to various tissue and cellular inputs. This adaptability is exemplified by the ER membrane, whose folding is modulated by the Lamin B receptor (LBR)[49]. The LBR forms part of a meshwork of intermediate filament proteins, creating a mechanical scaffold associated with adhesion receptors and the contractile cytoskeleton. Additional mechanisms, including focal adhesion-based and myosin II-dependent processes, may also contribute to ER membrane stretching. Notably, the ER membrane is continuous with the nuclear membrane, forming the nuclear envelope. This continuity suggests a potentially critical role for Piezo1 in mechanotransduction processes that span from the cell surface to the nucleus, further highlighting its importance in cellular mechanosensing and signaling pathways.

Piezo1's rapid Ca$^{2+}$ dynamics and short-lived cytoplasmic Ca$^{2+}$ release may serve as a potential trigger for Ca$^{2+}$-induced Ca$^{2+}$ release (CICR) through the RyR and IP3R channels[18,20,38,50]. Our surprising finding that IP3R expression is diminished in the SMCs of Piezo1$^{\Delta SMC}$ mice underscores a potential connection between ER-based Piezo1 and stretch- or compression-induced changes in SMCs[51]. Although others have found that IP3Rs may play a role in stretch-activated Ca$^{2+}$ release from the nucleus and ER, this process is likely tissue- and cell-type-dependent, emphasizing the need for further investigation in the context of SMCs.

Interestingly, the observed decrease in IP3R expression in Piezo1$^{\Delta SMC}$ mice suggests a complex interplay between Piezo1 and Ca$^{2+}$ signaling. This reduction, whether due to a feedback mechanism resulting from over-activation or secondary to loss of Piezo1, would lead to a relative depletion in cytoplasmic Ca$^{2+}$ but a relative abundance in ER/SR Ca$^{2+}$ stores. As a result, the requirement for store-operated Ca$^{2+}$ channels (Orai1 and Orai3) may be minimized[52], as suggested by their decreased expression in Piezo1$^{\Delta SMC}$ mice (Fig. 8; Supp. Figs. 5–6). The altered Ca$^{2+}$ distribution and IP3R expression may contribute to the reduced responsiveness to carbachol observed in the adult small bowel of Piezo1$^{\Delta SMC}$ mice without impacting period or duration (Fig. 9C–E). Notably, studies in endothelial cells have shown that Piezo1 activation induces Ca$^{2+}$ release through IP3R2, involving the generation of cAMP and subsequent IP3R2-evoked Ca$^{2+}$ gating[53]. Given the proximity of Piezo1 to SR/ER-anchored channels in SMCs, a similar pathway may be involved, potentially facilitating additional channel openings through cooperative gating[54]. However, further research is needed to establish this mechanism for Piezo1-mediated mechanotransduction in SMCs definitively.

In contrast, the murine pup IMC co-culture exhibits an alternative Ca$^{2+}$ signaling pathway that appears to be independent of Piezo1 in SMCs (Fig. 5). This finding underscores the developmental aspects of these signaling mechanisms and suggests that Piezo1's role in Ca$^{2+}$ signaling may evolve during intestinal maturation. These adaptive responses underscore the intricate interplay between various calcium channels and signaling pathways in maintaining Ca$^{2+}$ homeostasis without Piezo1.

While this study highlights significant insights into the role of Piezo1 in small bowel SMCs, several limitations should be acknowledged. First, the Myh11-CreERT model widely used restricted our knockout studies to male mice, necessitating future studies to fully address potential sex differences. Second, the expression of the Myh11 promoter in other organs and cell types may contribute to the

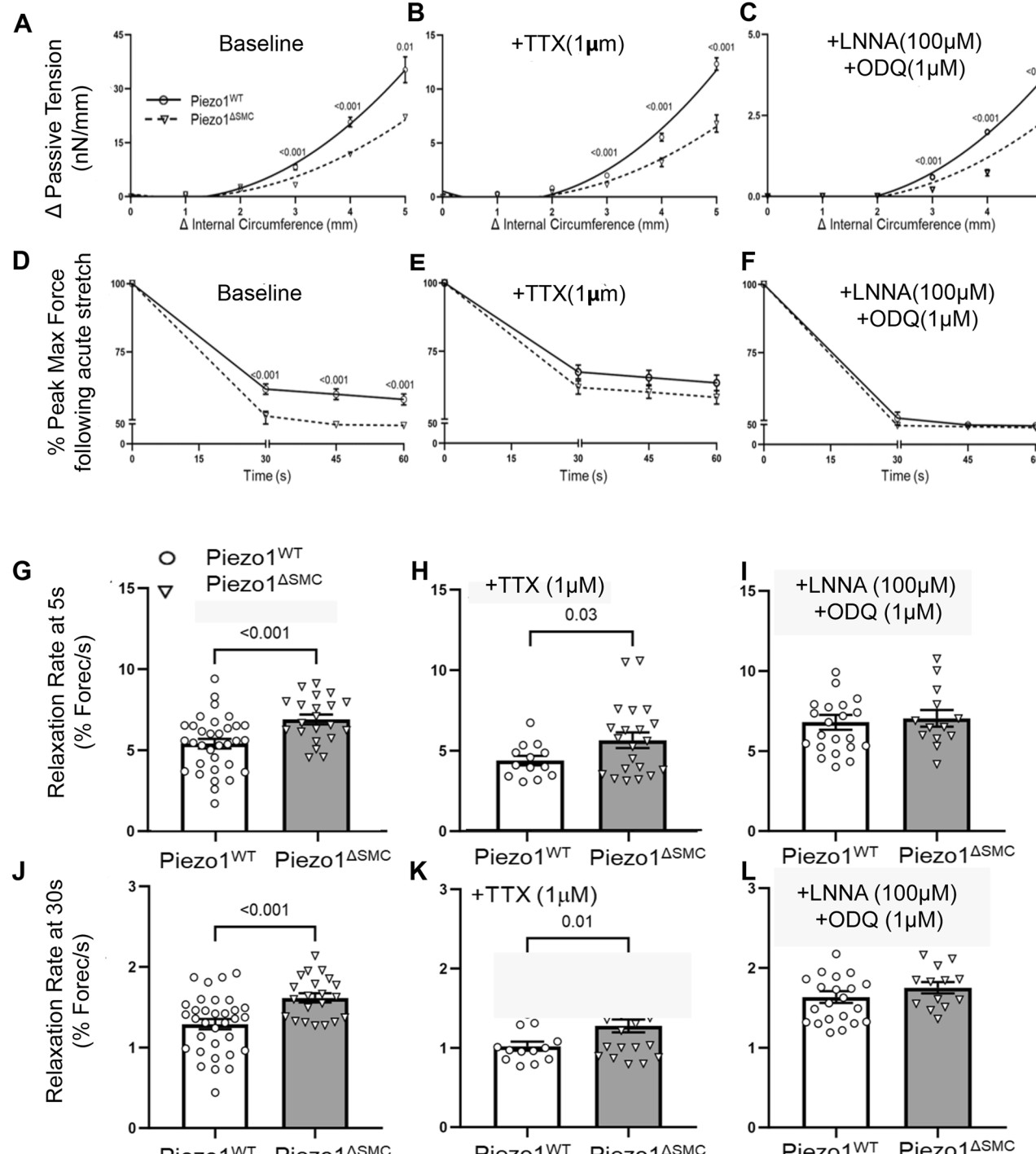

**Fig. 3 | SMC-specific Piezo1 depletion alters tonicity and tissue structure.** Length-tension relationships were assessed using passive tension following stretch at increasing circumferential stretches to determine tissue tonicity. Linear regression analysis with multiple t-tests showed (**A**) a reduction in the length-tension curve in Piezo1$^{\Delta SMC}$ bowel segments, which persisted in the presence of (**B**) TTX and (**C**) L-NNA/ODQ. Viscoelastic properties were assessed using percent reduction from max (peak) force (Fig. 2A, blue asterisks) over time until SS was reached. Compared to Piezo1$^{WT}$, Piezo1$^{\Delta SMC}$ bowel segments had (**D**) a greater reduction in max (peak) force at baseline conditions than in the presence of (**E**) TTX and (**F**) L-NNA/ODQ. ($N \geq 4$ mice per group, analyzed with simple linear regression and multiple t-tests with Bonferroni correction). The rate of relaxation from the max force was increased in Piezo1$^{\Delta SMC}$ bowel at (**G**) 5 s and (**J**) 30 s, which was retained with the addition of (**H**, **K**) TTX but not with (**I**, **L**) L-NNA/ODQ. ($N = 3$ mice per group were analyzed with Welch's t-test with multiple comparison tests.)

observed phenotype but is beyond the scope of this study, thus warranting further investigation. Third, our in vitro studies used muscularis isolated from 10-day-old pups, while contractility assessments were performed ex vivo in young adult mice. This age discrepancy limits direct correlations between measurements, as IMC excitability and function may be time and age-dependent. Fourth,

while we observed changes in calcium signaling and contractility, the precise molecular mechanisms linking Piezo1 to these processes remain to be fully elucidated. Advanced imaging techniques and real-time calcium measurements in vivo could offer more detailed insights into these dynamics. Future studies using complementary approaches could provide additional mechanistic insights and facilitate more

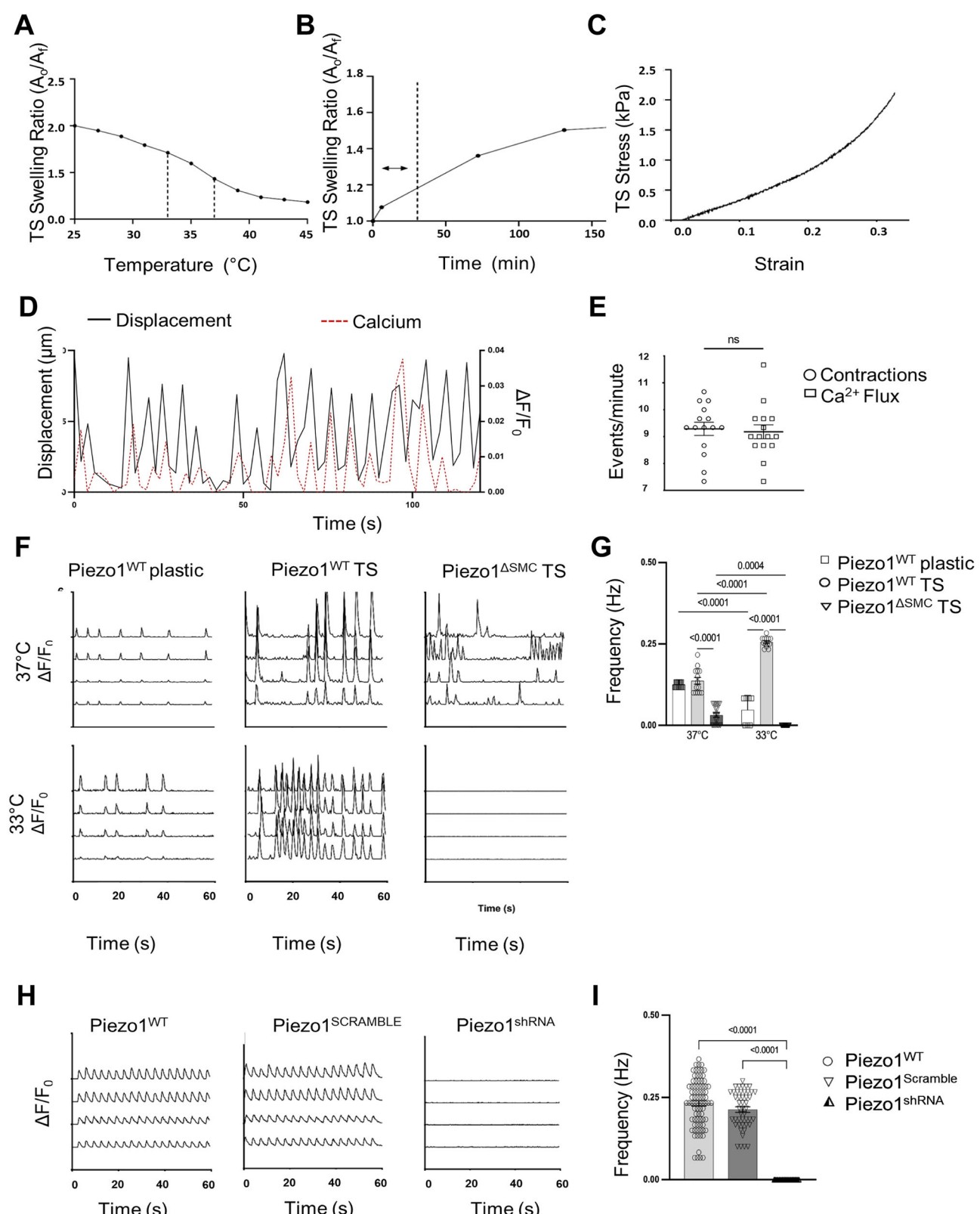

comprehensive comparisons with existing models. Lastly, our model focuses on SMC-specific Piezo1 deletion, which does not fully capture the complex interplay between different cell types in the intestinal wall. Future studies using cell-specific knockouts in other SIP syncytium components could provide a more comprehensive understanding of Piezo1's role in intestinal motility. Addressing these

limitations in future research will further enhance our understanding of Piezo1's critical role in intestinal physiology and potentially guide the development of targeted therapies for motility disorders.

In conclusion, our study reveals a critical role of Piezo1 in modulating small bowel SMC function and intestinal motility. The loss of Piezo1 in SMCs leads to significant alterations in gut transit, muscularis architecture,

**Fig. 4 | Thermosensitive stretch-inducible hydrogels with co-cultured IMCs showed altered Ca²⁺ signaling with loss of SMC Piezo1. A** Hydrogel properties showing a temperature range of 37 °C–33 °C selected for TS scaffolds to achieve optimal surface area change based on equilibrium swelling behavior. **B** TS scaffold swelling rate/ratio determined the time required to achieve stretching of IMCs in an isotropic manner, with a dashed line to mark the operation range of 30 min for a 20% area increase. **C** The stress-strain behavior of TS scaffolds (4.1 kPA) was modified to simulate the mechanical properties of small intestine tissue (4.1–4.5 kPA)[50]. **D** IMCs isolated from external muscularis strips from 8 to 10-day-old murine pups ($N = 6–8$ murine pups per biological sample) seeded on plastic or thermosensitive (TS) stretch-inducible hydrogels with spontaneous contractile behavior. Representative tracings depicting contractions (black) as measured by displacement (left axis) overlapped Ca²⁺ flux (red dashed) measured by absolute intensity changes, $\Delta F/F_0$

(right axis) with (**E**) frequency measurements of contractions and Ca²⁺ flux shown ($n > 12–15$ time points per group). **F** Representative GCaMP6f Ca²⁺ tracings ($\Delta F/F_0$) of Piezo$^{WT}$ IMCs seeded on plastic and TS hydrogels that stretch with a temperature reduction from 37 °C to 33 °C compared to Piezo1$^{\Delta SMC}$ IMCs seeded on TS. **G** Two-way ANOVA was used to measure change in frequency with change in temperature using Sidak correction. Significance values indicated in figure. Data displayed as mean ± SEM, $n > 6$ biological samples for each group (5–8 murine pups per biological sample). **H** IMCs isolated from muscularis layers of murine 10-day-old pups ($N = 6–8$ murine pups per biological sample). Representative GCaMP6f Ca²⁺ tracings ($\Delta F/F_0$) of Piezo1$^{WT}$, Piezo1$^{SCRAMBLE}$, and Piezo1$^{shRNA}$ IMCs seeded on a plastic scaffold at baseline, measuring (**I**) frequency differences ($n > 10$ biological samples per group, $N = 6–8$ murine pups per biological sample).

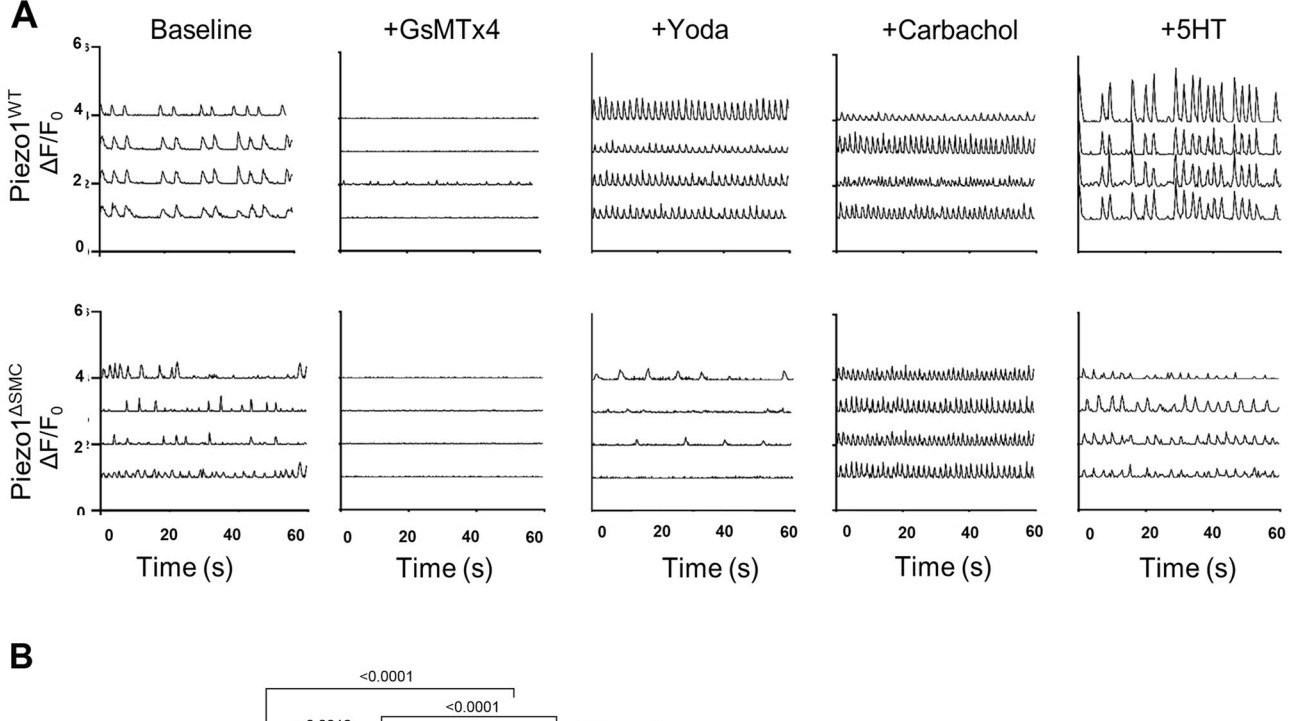

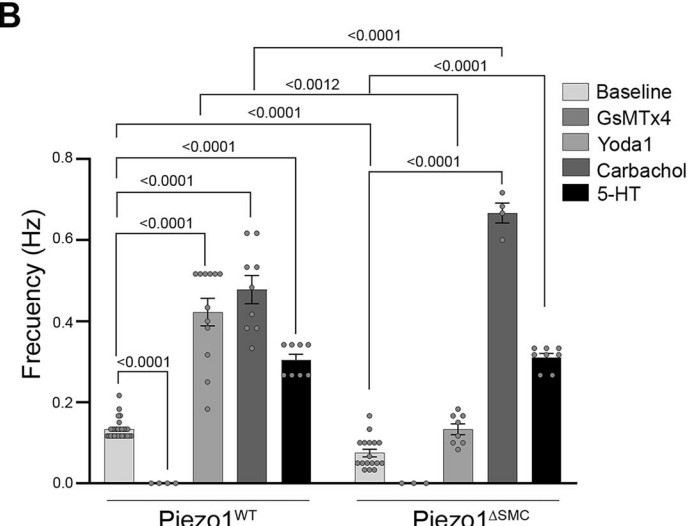

**Fig. 5 | Piezo1 modulators on in vitro Ca²⁺ flux.** IMCs were isolated from muscularis layers of murine 8-10-day-old pups (N = 6–8 murine pups per biological sample). **A** Representative GCaMP6f Ca²⁺ tracings ($\Delta F/F0$) of Piezo1$^{WT}$ and Piezo1$^{\Delta SMC}$ IMCs with GsMTx4 (20 μM), Yoda1 (5 μM), Carbachol (10 μM) and 5-HT (10 μM) between Piezo1$^{WT}$ and Piezo1$^{\Delta SMC}$ with (**B**) frequency differences between groups (n > 6 biological samples per group, N = 6–8 murine pups per biological sample) compared. Two-way ANOVA with interaction followed by Tukey's post-hoc analysis was applied.

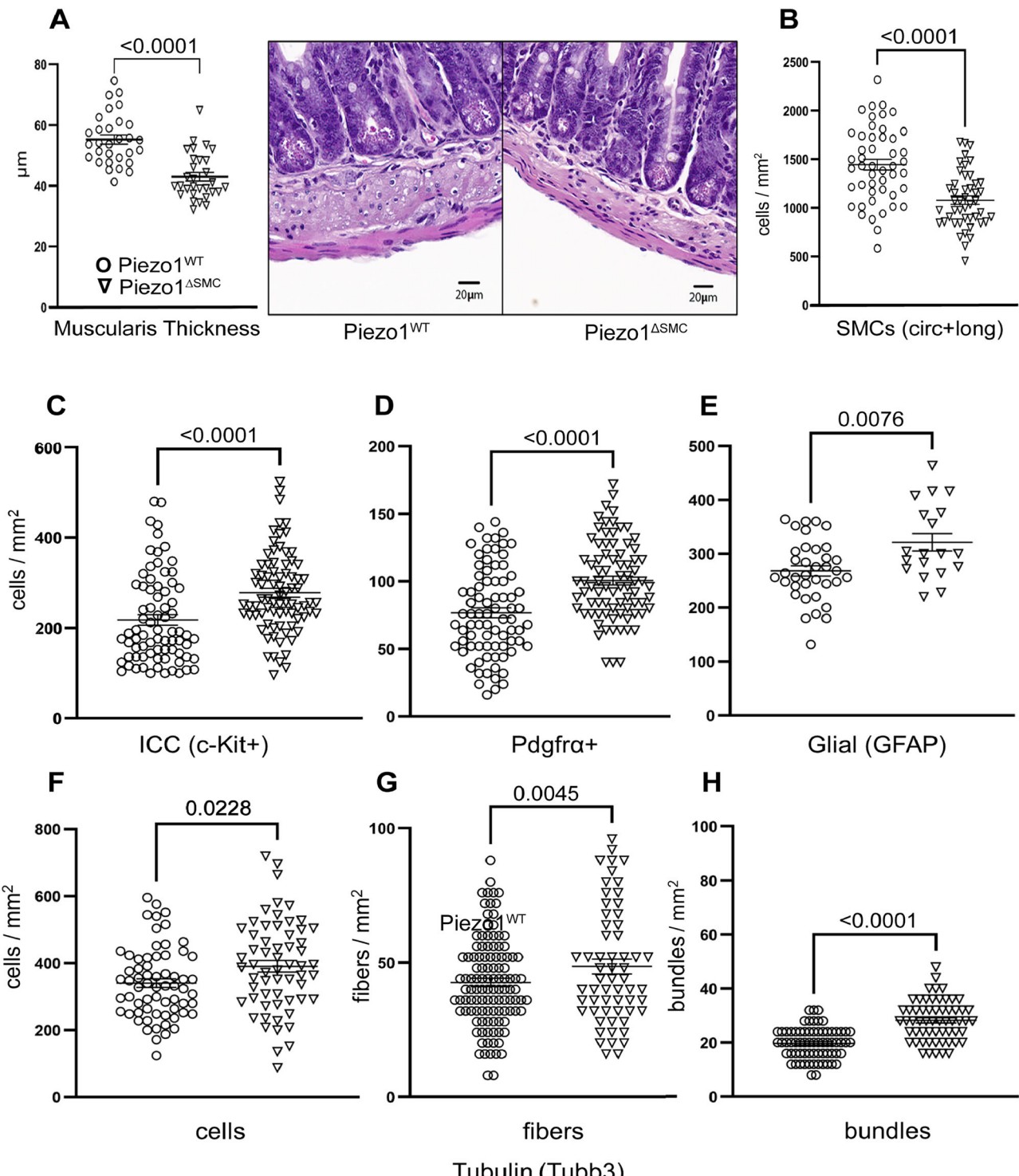

**Fig. 6 | Cellular composition of the muscularis layer following the loss of SMC Piezo1. A** Total muscularis thickness was assessed on H&E (right) stained FFPE samples from distal small bowel segments of Piezo1[WT] and Piezo1[ΔSMC] mice at 7–10 weeks old following post-Tam administration performed at 4–6 weeks old. 8–10 representative regions per mouse were measured from 3 to 4 mice per group. **B** The SMC density (SMCs per mm²) in circular and longitudinal layers in a WM preparation is decreased in Piezo1[ΔSMC] distal small bowel compared to Piezo1[WT] mice (N = 3–4 mice per group). **C–E** ICC, Pdgfrα + , and glial cell density as measured by staining for c-Kit (R&D Systems), Pdgfrα (Cell Signaling), and Gfap (Abcam) of WM preparations in Piezo1[ΔSMC] mice compared to Piezo1[WT]. Neuronal number of (**F**) cells, (**G**) fibers, and (**H**) bundle density per mm² as measured by staining for Tubb3 (Abcam) in Piezo1[ΔSMC] mice compared to Piezo1[WT] using WM preparations. Unpaired two-tailed t-test with Welch's correction. Data presented as mean ± SEM, n > 18 areas (N = 3–4 mice/group) with p-values indicated above, significance set at p < 0.05.

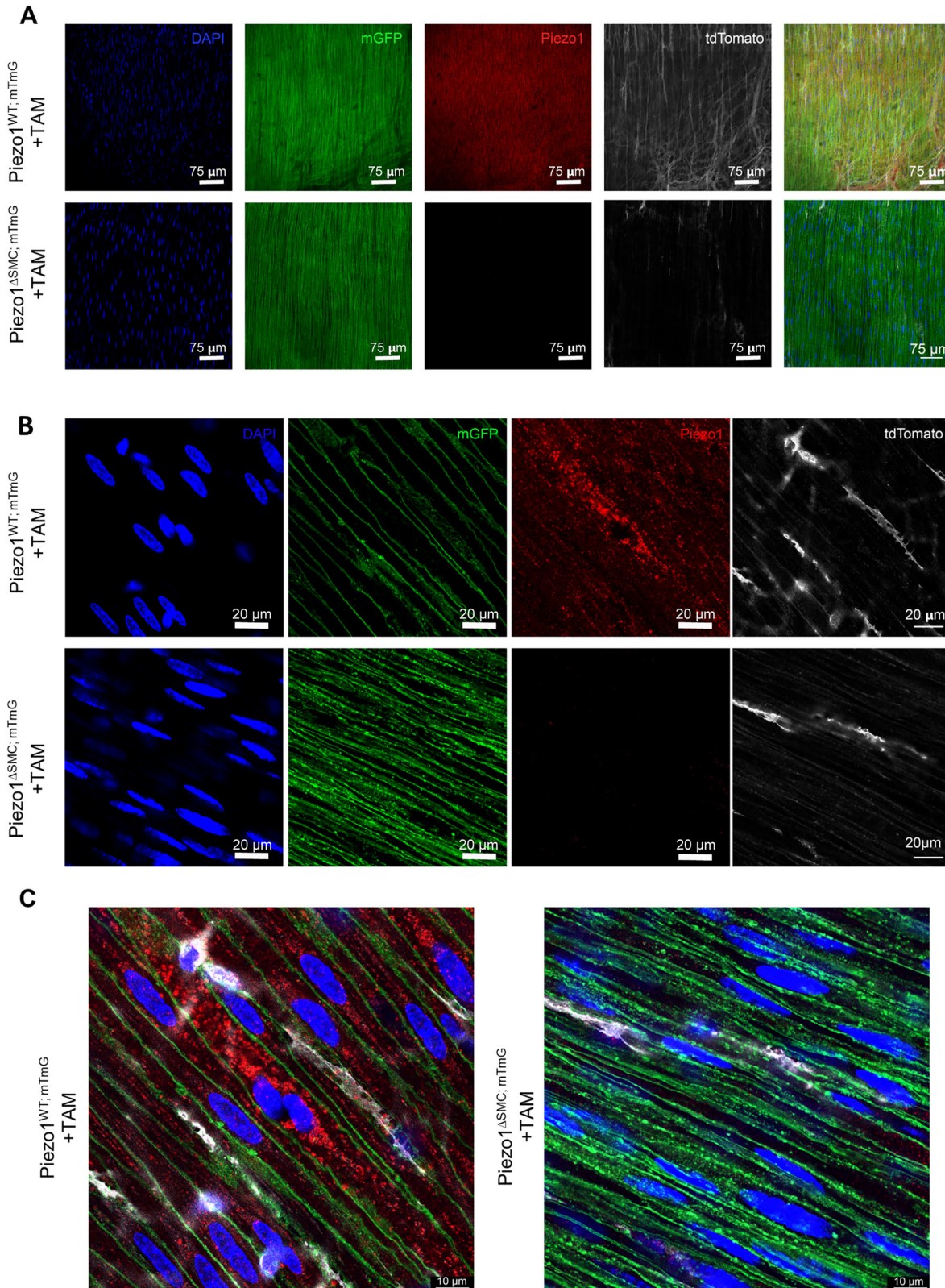

**Fig. 7 | Expression and localization of Piezo1 in small bowel muscularis. A–C** WM prepared distal bowel samples from Piezo1^(WT;mTmG) and Piezo1^(ΔSMC;mTmG) mice and 21 d post-Tam were used for IF staining of Piezo1 as imaged by Leica Stellaris 8 microscope (**A**), and Leica Confocal SP8-STED microscope (**B**, **C**). Green shows the membrane-GFP; pseudo-white is the membrane-Tdtomato from the mTmG reporter, and the anti-Piezo1 (Alamone Labs) signal is pseudo-red. The scale bars are (**A**), 75 μm; (**B**), 20 μm; and (**C**) 10 μm, respectively.

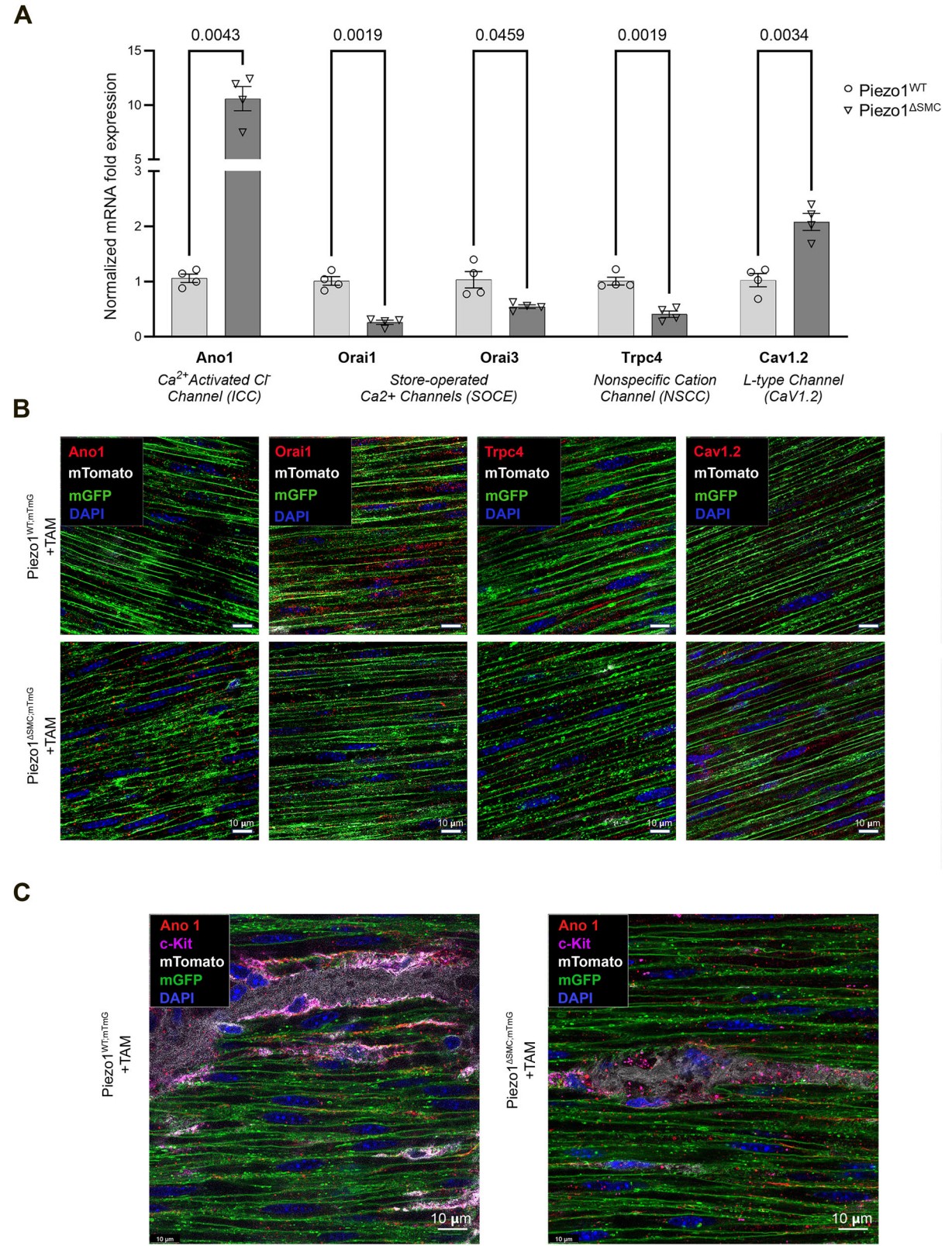

**Fig. 8 | SMC-specific deletion of Piezo1 alters related ion channel expression within the muscularis. A** Transcriptional expression (mRNA) of *Ano1*, *Orai1*, *Orai3*, *Trpc4*, and *Cav1.2* in Piezo1$^{\Delta SMC}$ and Piezo1$^{WT}$ mice from muscularis samples isolated from the distal small intestine. Normalized mRNA fold expression based on qRT-PCR results was calculated using the ΔΔCT method relative to *Gapdh* expression. Multiple t-test analyses with Welch's correction were performed with a False Discovery Rate (FDR) set to 5% with the corrected p-values (q-values) above comparison groups. Data are shown as the mean ± SEM (N = 4–5 mice per group). **B** WM distal small bowel samples from Piezo1$^{WT;mTmG}$ and Piezo1$^{\Delta SMC;mTmG}$ mice 21 d post-Tam were used for IF staining with a Leica Confocal SP8-STED microscope of various proteins required for Ca$^{+2}$ homeostasis. Green shows the membrane-GFP; pseudo-white is the membrane-Tdtomato from the mTmG reporter. Antibodies against Orai1, Trpc4, Cav1.2, and Ano1 (Alomone Labs) were pseudo-red. Scale bars, 10 μm. **C** Imaging of the muscularis from WM-prepared samples stained for Ano1 and c-Kit indicates Ano1 co-localization with c-Kit as expected in Piezo1$^{WT;mTmG}$ sample, and an increase of Ano1 staining along the GFP+ plasma membrane of SMCs in Piezo1$^{\Delta SMC;mTmG}$ mice.

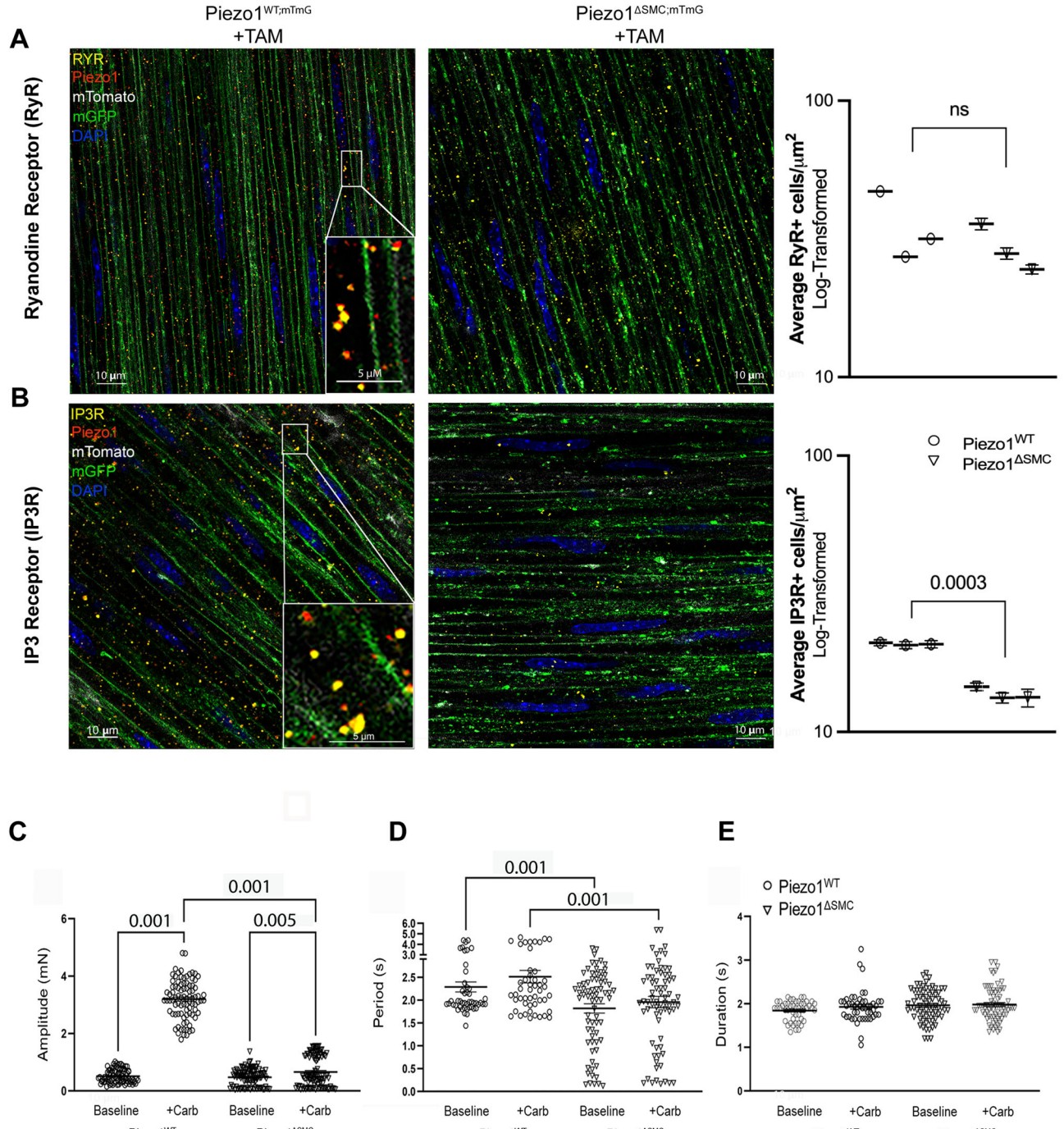

**Fig. 9 | Piezo1 protein in SMCs colocalizes with IP3R and RyR.** WM sample of distal small bowel isolated from Piezo1$^{WT;mTmG}$ and Piezo1$^{ΔSMC;mTmG}$ mice 21 d post-Tam, were imaged with Leica Confocal SP8-STED microscope. **A** The anti-RyR (Santa Cruz) signal is pseudo-yellow, the anti-Piezo1 (Alamone Labs) signal is pseudo-red, green shows the membrane-GFP, and pseudo-white is the membrane-Tdtomato from the mTmG reporter. All scale bars are at 10 μm and insert at 5 μm. **B** The anti-IP3R (BD Bioscience) signal is pseudo-yellow, while the anti-Piezo1 antibody and other signals from the mTmG reporter are identical to A. Immunofluorescence-stained IP3 and RYR protein channels were quantified using AIVIA software Pixel Classifier and Smart Segmentation tools to segment structures, distinguish them from the background, and generate 2D outlines. Total counts were then normalized by ROI area relative to sample data to account for batch and animal differences ($n = 25$ ROIs per mouse, $N = 3$ mice per group). Data shown as mean +/− SEM. Statistical analysis was performed on log-transformed averages per mouse per group with significant p-values indicated above, with full nested analysis included in supplementary fig. 8. **C, D** Isometric force assessments on intact, full-thickness bowel segments from Piezo1$^{ΔSMC}$ mice have impaired response to carbachol at SS compared to Piezo1$^{WT}$, including amplitude, (**D**) period, and (**E**) duration ($n ≥ 20$ measurements from N = 3 mice per group).

contractility, and ion channel expression, without causing severe morbidity. This suggests a nuanced role for Piezo1 in intestinal homeostasis. The predominant localization of Piezo1 within subcellular regions, particularly its colocalization with RyR and IP3R, underscores its potential role in regulating intracellular $Ca^{2+}$ dynamics. Our findings suggest that Piezo1 may trigger CICR in response to mechanical cues, potentially facilitating cooperative gating and signal amplification, thereby contributing to SMC receptiveness and function within the SIP syncytium. Ultimately, this work

emphasizes the complex and essential role of Piezo1 in maintaining intestinal motility and function, underscoring the importance of mechanosensitive ion channels in gastrointestinal physiology.

However, further investigation is essential to fully comprehend Piezo1's complex role in maintaining intestinal motility and function. Future studies may focus on elucidating the dynamics of Piezo1's subcellular movement and its intricate interactions with other ion channels in SMCs[55]. This deeper understanding will not only provide crucial insights into the physiological function of Piezo1 but may also unveil potential therapeutic targets for gastrointestinal disorders associated with impaired SMC function or calcium signaling abnormalities.

## Methods
### Murine crosses and breeding
Mice carrying the Piezo1[fl/fl] gene obtained from *the Jackson Laboratory* were bred with mice carrying the Myh11-Cre[ERT] gene to generate Myh11-Cre[ERT]: Piezo1[fl/fl]. *mTmG* mice constitutively expressing Tdtomato at baseline and GFP with Cre-activation were crossed with Myh11-Cre[ERT]:Piezo1[fl/fl].[29] All mice were ordered from *Jackson Laboratory*, as indicated by (Supplementary Table 4). Wildtype (WT), Piezo1[WT/WT]/Myh11[ERT2/Cre] (Piezo1[WT]), and homozygote knockout (KO), Piezo1[fl/fl]/Myh11[ERT2/Cre] (Piezo1[ΔSMC]) were generated[28]. To specify cells with Cre induction, mice were generated with mTmG reporter (Piezo1[WT;mTmG] and Piezo1[ΔSMC;mTmG])[29].

Mice were kept under constant temperature conditions (23 ± 2 °C) with standard chow, water ad libitum, and a 12-h light/dark cycle. Mice were genotyped by PCR and gel electrophoresis. All experiments were approved by the UCLA Institutional Animal Care and Use Committee (IACUC) protocol # ARC-1996-070-TR-002.

### ARRIVE guideline information
We have complied with all relevant ethical regulations for animal use. During the breeding process, the mice were fed irradiated 5053 PicoLab Rodent Diet 20. Mice were in ventilated Lab Product cages and provided access to water via an Automatic Water System. Corn Cob bedding supplemented with Nestlets was used, and cages were changed bi-weekly. The Biosecurity level was A2 Silver with a light cycle of 6 am–6 pm. For tamoxifen administration, mice were moved to a biohazard mouse facility and fed non-irradiated 7013 NIH-31 Mouse/Rat Sterilizable Diet by ENVIGO (Harlan Teklad). Mice were maintained in disposable Innovive cages and given pre-filled disposable water bottles with acidified water by Innovive. The bedding included corn cob from Innovive and contained pre-inserted 100% virgin kraft nesting enrichment sheets from Innovive. The Biosecurity level was ABSL 2 with a light cycle of 6 am–6 pm.

### Tissue Preparation
Mice were euthanized by $CO_2$ inhalation, followed by cervical dislocation. The small intestine was then isolated and rinsed with PBS. Whole tissue samples, isolated muscularis externa[21], were obtained for molecular analysis using established methods.

### Genetic Knockout Confirmation
DNA was isolated from tissue and culture samples using the Invitrogen PureLink Genomic DNA Mini Kit. PCR was performed to confirm KO using previously described methods[30]. Protein was extracted from tissue and boiled in Laemmli buffer, and 10 μg was loaded onto 4–15% polyacrylamide gel, separated by SDS-PAGE, and transferred onto a PVDF membrane (GE Healthcare). We detected by Western blot analysis using the appropriate antibody and ECL plus western blotting kit (Thermo Scientific).

### RNA Extraction and RT-qPCR analysis
RNA was isolated from the tissue and in vitro samples using the Invitrogen PureLink RNA Mini Kit. RT-qPCR was performed using Thermo Fisher 1-Step Taqman Master Mix and Taqman Probes for the following genes: Piezo1, Piezo2, Ano1, Trcp4, Orai1, Orai3, Cav1.2, and Gapdh. The mRNA

data were analyzed using the ΔΔCT method and normalized to *Gapdh* expression (Supplementary Table 2).

### Formalin-Fixed Paraffin-Embedded and Flash-Frozen Immuno-fluorescence Staining
The tissue was either FFPE or flash-frozen, depending on the antibody specifications. FFPE samples were placed in formalin for 24 h and then transferred into 70% ethanol. FFPE tissues were deparaffinized with xylene. After two 4-min xylene soaks, followed by two 5-minute soaks in 100% ethanol, two 3-min soaks in 95% ethanol, one 3-min soak in 80% ethanol, one 5-min soak in 70% ethanol, and one 1-min soak in $H_2O$ in that order. The tissue was then treated with 0.3% $H_2O_2$ for 15 min. Next, heat-induced epitope retrieval was performed with 10 mM Tris 1 mM EDTA Buffer at pH 9. Tissue was then permeabilized in 0.2% Triton X-100. Blocking was performed using Dako Protein Block with 0.05% Triton. Primary antibodies were incubated overnight at 4 °C (Supplementary Table 1). Slides were then incubated with the secondary antibody for 45 min at RT, followed by nuclear staining with Hoechst stain for 15 min. Sections were also stained for hematoxylin and eosin (H&E).

### Whole-mount Tissue Processing and Staining
The small intestine was dissected and cleaned, and distal ileal segments measuring 2.0 cm were obtained. Longitudinal incisions were made, and the tissue was pinned to silicone plates, fixed in 4% paraformaldehyde in PBS pH 7.4 overnight at 4 °C, and then washed with PBS. Tissue strips of 5 mm thickness were cut using a dissecting microscope[56]. These square pieces were carefully rinsed in PBS and permeabilized in 1% Triton-X100 in PBS for 1 h. Blocking was performed using a Dako Protein Block with 0.05% Triton for 1 h at room temperature. Primary antibodies were incubated for two nights at 4 °C, and secondary antibodies were incubated overnight at 4 °C, followed by staining with Hoescht (Supplementary Table 1). For Ano1, Orai1, Trpc4, and Cav1.2, antibody staining of WM samples was performed as stated below, followed by Hoescht counterstaining for 15 min (Supplementary Table 1). For the co-staining of RYR(F-1) with Piezo1 and IP3R3 with Piezo1, the exact conditions were met by adding a second primary antibody. The secondary antibody was incubated with anti-mouse 790 and anti-rabbit 647, followed by Hoescht counterstaining for 15 min (Supplementary Table 1).

### Imaging and quantification of immunostained cells
Imaging was performed with a Leica Stellaris 8 microscope, Zeiss Confocal LSM880, and the Leica Confocal SP8-STED/FLIM/FCSSP8 microscopes. Positive marker antibody immunostaining was defined as cells with increased fluorescence from the background correlated with nuclear (Hoechst). Smooth muscle cells (GFP), Neuronal (Tubb3), glial (Gfap), interstitial cells of Cajal (c-Kit), and Pdgfrα+ cells (Pdgfrα) were quantified using WM preparations for improved accuracy[57,58]. Neuronal fiber density (Tubb3) was further quantified using the number of fiber bundles or single fibers that crossed the 0.5 × 0.5 mm grid superimposed as described[57].

### Quantification of immunostained RyR and IP3R clusters
Images were analyzed using AIVIA software (DRVISION Technologies) to quantify immunofluorescence-stained protein channels. The Pixel Classifier tool was used to enhance specific features within the images, with representative structures manually selected to train the system for classification. Using the Smart Segmentation feature, the software applied these training parameters to detect and segment structures across the entire image, generating outlines in 2D, and segmented structures were quantified. The data were then exported for further analysis in R, where additional computations and ROI generation were performed to refine the evaluation of spatial distribution. R script for RYR-IP3 structure generated for batching count processing[59].

### Hydrogel Scaffold Synthesis, Preparation, and Characterization
The TS scaffolds were N-isopropylacrylamide (2210-25-5), 273 mg/mL acrylamide (79-06-1), 27 mg/mL, copolymer poly(Nipam-co-Aam). The

crosslinker and photo-initiator used were N,N'-Methylenebisacrylamide (110-26-9), 10 mg/mL, and 2-Hydroxy-2- methylpropiophenone (7473-98-5), 10 uL/mL. All reagents were dissolved in dimethyl sulfoxide (DMSO). 100uL of the solution was pipetted into a PDMS elastomer mold and polymerized under UV light (15 s). All chemicals purchased from Sigma. Collagen Type I and polydopamine coating were introduced to the hydrogel to improve biocompatibility and cell adhesion. Collagen Type I (CAS #9007-34-5) and Dopamine Hydrochloride (CAS #62-31-7) were purchased from Sigma and Fisher Scientific, respectively. TS hydrogels were immersed in 33% collagen solution (diluted in DI water). Hydrogels were exposed to heating/cooling cycles for efficient collagen absorption by placing them in hot (40 °C) and cold-water baths (-5 °C) with 3-min intervals for 4 cycles. Hydrogels were transferred to a dopamine solution (150 mg/ 50 mL in DI water) for 1 min before 17 mL of 0.2 M pH 8.6 buffer (0.2 M disodium phosphate and 0.2 M monopotassium phosphate mixed at a 9.9:0.1 ratio) was added to increase the pH and initiate polymerization. The reaction was terminated after 90 min by transferring the hydrogels to DI water. The swelling behavior, swelling rate, and Lower Critical Solution Temperature were determined by capturing hydrogel images at different points and measuring their dimensions using ImageJ. Mechanical properties were characterized using a uniaxial testing machine (Dynamic Mechanical Analyzer, from TA Instruments), and the modulus was calculated using the linear segment of the curve[60].

## Murine Pup IMC culture

Piezo1[WT] and Piezo1[ΔSMC] mouse pups were sacrificed at 10 d of age by $CO_2$ inhalation, followed by cervical dislocation. Muscularis samples isolated from the entire length of the small bowel were used and cultured as previously validated in our lab[21]. Piezo1 was knocked out by treating muscularis cells with 0.1 μM 4-OHT for 24 h or with Piezo1 shRNA lentivirus for 24 h. $Ca^{2+}$ influx was visualized by transducing with lentivirus expressing gCaMP6f for 24 h. Chemical reagents were added at the following concentrations: GSMTx4 (20 μM), Yoda1 (5 μM), 5-HT (10 μM), and Carbachol (10 μM) (Supplementary Table 5).

## Human IMC culture

All human intestinal samples were obtained from de-identified and discarded fetal tissue specimens following pathology evaluation by the UCLA translational pathology core through an IRB approved by the UCLA Institutional Review Board (IRB #11-002504). Human muscularis was extracted, cultured as described above, and treated with GSMTx4 (20 μM) and Yoda1 (5 μM). All other steps were identical to the murine IMCs described above[21].

## In vitro Ca$^{2+}$ Flux/Contractility Analysis

IMCs were visualized using a Zeiss Spinning Disc confocal microscope. Videos were recorded for up to 3 min, and regions of interest (ROIs) of $Ca^{2+}$ flux were obtained using *Zen Blue 3.3* software. For each 1–3 min recorded video, we observed no fluorescence cells drifting in or out of the frame to confirm that scaffold movement did not contaminate the capture of $Ca^{2+}$ signaling through the ROIs. A custom *MATLAB* code[61] was used to process raw data and count peaks (events) by identifying a baseline and data points that were significantly higher than the baseline (peak data points) and those that were very close (peak base data points). Variables like frequency could then be calculated. Data were subsequently exported to GraphPad Prism for analysis. Graphs were plotted using absolute intensity differences over the baseline intensity (ΔF/F0)[62,63].

## Whole-bowel transit time

5% carmine-red dye and 1.5% methylcellulose in water were given to Piezo1[WT] and Piezo1[ΔSMC] mice via orogastric gavage. After administration, the mice were separated into clean individual cages with free access to food and water. Each cage was checked at 15-minute intervals for stool color and number for 8 h. The time to first red stool was defined as the total transit time up to 8 h max.

## Wire Myography

Piezo1[WT] and Piezo1[ΔSMC] adult mice were euthanized by $CO_2$ inhalation and cervical dislocation immediately before experiments. Distal ileum segments were carefully separated from the mesentery, dissected, cleaned, and cut into 3 mm rings. Each ring was mounted on a wire myograph per protocol (Danish Myo Technology, Aarhus, Denmark). Preparations were kept in physiological saline solution [119 mM NaCl, 4.7 mM KCl, 1.2 mM $MgSO_4 \cdot 7H_2O$, 1.2 mM $KH_2PO_4$, 25 mM $NaHCO_3$, 6.1 mM $C_6H_{12}O_6$, 30 μM EDTA, and 2.5 mM $CaCl_2 \cdot 2H_2O$], pH 7.4 at 37 °C. Preparations were allowed to equilibrate for 20 minutes, adjusting for tissue differences and baseline contractility and establishing a baseline tone of 2mN. Stretch experiments consisted of increasing diameter by 0.5 mm with each complete turn. Chemical reagents were directly added to the organ bath at the following final concentrations: Nicardipine (10 μM), GsMTx4 (1 μM), Carbachol (1 μM), LNNA (100 μM), ODQ (10 μM), and TTX (1 μM) (Supplementary Table 5). *LabChart software*, version 8, was used to obtain the following data from each experiment: amplitude, period, duration, and integral force. For experiments measuring the effect of each inhibitor, the baseline amplitude was obtained at 2-min intervals before the addition of the inhibitor. The amplitude of subsequent contractions following the addition of the inhibitor was obtained and converted to relative values compared to the baseline of the same tissue to normalize the data across different tissues.

## Statistics and Reproducibility

Statistical analyses were performed using *the GraphPad Prism* software. Unpaired Students' *t*-tests with Welch's correction were used for two group comparisons. Multiple t-test analysis with a false-discovery rate (FDR) set to 5% was applied for multiple comparisons of unrelated groups. Two-way ANOVA with main effects for Piezo1 and stretch and their interaction was used for hypothesis testing with post-hoc analyses, as specified. Nested analysis performed to assess means between groups to consider any variation between mice due to animal or batch differences. General linear regression models with main effects were used as independent variables to estimate effect sizes (adjusted mean differences). Statistical significance was set at $p < 0.05$. Descriptive statistics were obtained when appropriate to calculate the coefficients of variation and distribution curves. Data shown as mean±SEM unless otherwise stated.

## Reporting summary

Further information on research design is available in the Nature Portfolio Reporting Summary linked to this article.

## Data availability

Analyzed data and detailed statistical analyses available in **Supplementary Data** available via Figshare (https://doi.org/10.6084/m9.figshare.28210808.v2). Any additional data desired is available upon request.

## Code availability

RR script for RYR-IP3 structure batching count processing can be accessed via Github: https://github.com/NRKushnir/RYR-IP3-Structure-Count-Batch-Processing and Zenodo (https://doi.org/10.5281/zenodo.14636293)[59]. This script processes data exported from AIVIA software to analyze IP3R and RYR protein structures in murine smooth muscle cells, organizing data into regions of interest (ROIs) within a 5 × 5 grid. It generates multiple output files summarizing the structure counts, ROI areas, and positional data to streamline analysis for wild-type (WT) and knockout (KO) datasets. MATLAB code used to analyze $Ca^{2+}$ influx curves can be accessed via GitHub (https://github.com/ydu1955/Piezo1_mSMC_Calcium-Influx/tree/MATLAB_Piezo1_YD) and Zenodo (https://doi.org/10.5281/zenodo.14722519)[61,64,65]. The key operations of the code are identifying and isolating mSMC calcium influx signal peaks to analyze their intensity change, peak duration, frequency, and magnitude. The code parameters can be adjusted in order to account for additional background noise that may be included with the calcium signals (see code commentary

and ReadMe document). The output files summarize the listed variables and data required to graph the calcium behavior.

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

## Acknowledgements

We sincerely thank Sal Baker, Ph.D. (University of Nevada), for his extensive guidance in evaluating our calcium signaling data, and Daniel J. Tancredi (University of California, Davis) for his support in choosing the appropriate statistical tests to rigorously assess our data. We would also like to thank Ivette Perez, Ariana Infant, Margaret Kim, Alex Nguyen, Evelyn Arambula, Andrew Rodriguez, and Diana Mena (University of California, Los Angeles; California State University, Northridge) for their assistance with the various experiments. We also acknowledge the funding agencies supporting the various grants used to perform this research: the National Institute of Diabetes and Digestive and Kidney Diseases (NIDDK) (T32 DK007180, R01 DK083762, P30 DK41301, RC2 DK118640, and R01 DK132319), National Center for Advancing Translational Sciences (NCAT) (KL2 TR001859), the Children's Discovery Institute at UCLA (CDI-FRSA-07012019), The Hartwell Foundation, the American Academy of Pediatrics Marshall Klaus Research Program (20202900), and the California Institute for Regenerative Medicine (TB1-01183).

## Author contributions

Conceived and designed the experiments: G.B., Y.D., X.H., M.S., J.D., M.G.M. Performed the experiments: G.B., Y.D., N.S., M.M., A.F., N.K., N.P.N., E.T., R.S.S.V. Analyzed the data: G.B., Y.D., N.S., M.M., A.F., E.T., R.S.S.V., N.K., M.S., J.D., M.G.M. Contributed reagents/materials/analysis tools: M.L., X.H., M.S., J.D., M.G.M. Wrote the paper: G.B., Y.D., N.S., M.G.M. Critical review of the manuscript: G.B., E.T., R.S.S.V., X.H., M.S., J.D., M.G.M.

## Competing interests

None of the authors have any potential financial, professional, or personal conflicts.
