## [Transparent Peer Review file · Communications Biology]

Smooth muscle cell Piezo1 depletion results in impaired contractile properties in murine small bowel

Corresponding Author: Dr Martin Martin

This manuscript has been previously reviewed at another journal. This document only contains information relating to versions considered at Communications Biology.

Version 0:

Reviewer comments:

Reviewer #1

(Remarks to the Author)

Brief summary of the manuscript: This paper describes the phenotype caused by loss of Piezo1 in bowel smooth muscle with a lot of detailed information about effects on physiology in vitro and in vivo including force measurements, in vivo transit times, contractile responses with or without various inhibitors, and nice imaging showing Piezo1 in an intracellular compartment that appears to be sarcoplasmic reticulum.

Overall impression of the work: This is a valuable contribution to the literature. All my comments can be addressed by careful revisions to the text, with a few minor exceptions. In some cases, there appear to be missing data (e.g., about IP3R abundance) or missing experimental details (e.g., about bowel obstruction studies). Figure legends in particular need careful proofreading.

As prior reviewers pointed out, there remain unanswered questions, but this is true about all manuscripts. Many of the “next level” studies are not trivial and could take months or years to complete, and apparently will be conducted by Dr. Bautista as she begins her independent scientific career. I agree with authors that these questions are best left for a separate manuscript. Those questions might include:

- Determining how Piezo1 impacts calcium release from sarcoplasmic reticulum compared to calcium entry via the plasma membrane.
- Determining how Piezo1 impacts movement of ions through other channels in various cellular domains.
- Defining how other cell populations respond to Piezo1 loss in smooth muscle and defining mechanisms behind these changes in cell biology.
- Determining why Ano1 levels change in Piezo1 conditional KO (cKO) mice.

As with any good paper, the new observations in this paper lead to many new questions. This is part of what makes the work interesting and valuable. I think these issues are best addressed in future manuscripts.

From a clinical perspective, this work is also valuable since mechanisms underlying myopathic chronic intestinal pseudo-obstruction remain incompletely understood, and problems with bowel motility (slow transit constipation, irritable bowel syndrome) are common and remain challenging to treat. The newly defined roles for Piezo1 in the bowel might be useful therapeutic targets for these and other disorders.

Specific comments, with recommendations to address each comment. There are many minor issues to address:

Line 112: “modulates” instead of “modulating”.

Line 115: “SIPgenic innervation” is an odd term that I cannot easily find referenced. I say “odd” because innervation, by definition, is something nerves do, and none of the SIP syncytium cells are nerves. I guess this is OK, but it might be more confusing than helpful for readers who think of “innervation” the way that I do.

Line 175: Says Piezo2 mRNA is reduced in bladder, but the graph shows Piezo 2 mRNA is slightly increase in the Piezo

cKO mouse. I am confused.

Line 206: Text says, "Piezo1 modulates the strength and pattern of contractions in response to mechanical forces, likely by contributing to the Cav1.2-mediated Ca²⁺ influx that triggers contractility." This is confusing. Data show loss of Piezo1 alters strength and patterns of contractions, but data in the preceding paragraph do not show calcium transients. Also, how might Piezo1 "contribute to Cav1.2-mediated Ca²⁺ influx"? Unless authors wish to add supporting data, I recommend omitting this speculation from the Results section.

Lines 212-214: Text says, "The addition of tetrodotoxin (TTX), an inhibitor of sodium channels essential for the generation of action potentials in neurons, and L-NNA, an antagonist of nNOS combined with ODQ, a soluble guanylyl cyclase (sGC) inhibitor⁶ to isolate EN modulation, did not impact the length-tension relationship in Piezo1 Δ SMC mice (Fig.3B/C)." This does not appear to be true. While the graphs in Figures 3A-C appear similar, the scale bars differ. I am also confused about why blocking neural signaling with TTX or blocking NO signaling in Fig 3C should reduce passive tension. Since NO is a major smooth muscle relaxant, I expected blocking NO (or blocking all nerve activity) to increase muscle tension. Is there a simple way for readers to understand these data?

Lines 251-252: Text says, "Overall, these data show that in an isolated in vitro model of IMCs, loss of Piezo1 in SMCs results in a disruption in Ca²⁺ flux, likely related to disruptions in the electrical coupling within the SIP syncytium". While they show loss of Piezo1 in SMC impacts calcium flux, I do not see evident of "disruption in electrical coupling within the SIP syncytium", which would presumably mean disruption of gap junctions, a topic for which there are no data. Please rephrase to avoid speculating far beyond the data, or provide data that electrical coupling is disrupted in the SIP syncytium after Piezo1 loss.

Lines 259-260: Text says, "We used the mTmG reporter mice to assess whether SMC depletion of Piezo1 SMCs in the muscularis following Tam." This seems like an incomplete thought. Please review.

Figure 8C: It would help to see the individual colors as separate panels, as authors did for other figures.

Line 304: Text says, "there was a significant reduction in IP3R expression in Piezo1 Δ SMC;mTmG compared to that in Piezo1WT;mTmG SMCs." Where are the data about IP3R abundance? To support this statement, I expected a figure to show qPCR data or quantitative immunohistochemistry. This seems like an important omission since authors again comment on reduced IP3R expression on lines 322, 376 and 381.

Lines 359-361: Text says, "While current models have predominantly placed Piezo1 primarily in the plasma membrane, supports its presence within subcellular organelles in other mechanically active organs." This sentence seems to be missing one or more words.

Line 376: Text says, "Our novel finding that IP3Rs in the ER are Piezo1-dependent". What are the data to support this statement?

Line 490: Text says, "Protein was extracted from tissue and lysed". What does it mean for protein to be "lysed"? I suspect authors mean "Protein was extracted from tissue and boiled in Laemmli buffer". Please clarify.

Line 502: Text says, "Paraffin-embedded tissues were deparaffinized on a heat block at 50°C for 10 min, then quickly transferred into a container containing xylene". The phrase "container containing" seems redundant. Also, it would be more accurate to say "deparaffinize with xylene" since heating to 50C does not remove paraffin.

Lines 505-506: Text says, "quenched in 0.3% H₂O₂". Would it be more accurate to say, "oxidized with" or "treated with", since it is not clear what hydrogen peroxide would "quench" in this context.

Line 529: Consider changing "interstitial" to "interstitial cells of Cajal", a specific cell type.

Lines 845-849: Text says, "(A) Myh11-Cre expression confirmed with Leica demonstrating distal small bowel segment from Piezo1WT mice containing the membrane-associated Tdtomato fluorescence (mTmG) reporter (Piezo1WT;mTmG) revealed Tdtomato fluorescence in all cell types across the layers of the bowel, and no membrane-associated GFP fluorescence (pseudo-green), indicating an absence of Cre expression without tamoxifen induction." This is confusing. What does "confirmed with Leica" mean? What are each of the 4 panels intended to show? Consider adding labels across the top to indicate what color was imaged. How is "Myh11-Cre expression confirmed" in panel (A) in the absence of added tamoxifen? Mouse line name is also inaccurate since this is "Myh11-ERT2/Cre". How old are these mice?

Lines 849 to 852: Text says, "(B) In contrast, 21d post-Tam treated Piezo1WT;mTmG mice, the epithelial layer retained Tdtomato, while the cellular components of the muscularis were either Tdtomato or GFP+." Were these mice the same age as the mice in panel (A)? Also, should say "In contrast, IN 21d post-Tam treated . . ."

More care should be taken with figure legends. I may not have identified all problems. Some issues are noted below:

Figure 1C: What age were the mice? How long after tamoxifen were analyses done?

Figure 1F: Was Carmine dye given "orally" as stated (which usually means "by mouth") or was Carmine given by gavage

(into the stomach).

Figure 2A in the green box: The “i” (amplitude) is not labeled on the figure. A “v” is labeled on the figure but the meaning of this “v” is not described in the figure legend. Also, black and green tracings are difficult to distinguish. Please consider adding labels or using colors that are more easily distinguished, including by color blind individuals. Parts of Figure 2A outside the green box are not described.

Figure 2 Legend: Text says, “Phasic contractile activity, including (B) amplitude, (C) AUC, (D) duration, and (E) period, were altered in Piezo1 Δ SMC mice compared to controls (n=50 samples, N=3 mice per group).” This is confusing. Did authors measure >15 ileal rings (3mm long each) per mouse? How is this possible? Alternatively, were 3 ileal rings evaluated per genotype after stretching each piece > 15 times?

Figure 3A-C: How is “internal circumference” measured while the tissue is in a pressure transducing apparatus?

Figure 3G-L: Y axis labels say “%Forec/s”. Should this say “%Force/s”? If so, the error was made twice.

Figure 3I and 3L: In the PDF version of the figures, labels above the graphs overlap with data points and the “ODQ (1 μ M)” is not rendering properly. Also, is there a reason for not analyzing at least 3 mice per group for Figure 3G and 3J?

Figure 6 Legend title: Text says, “Gross and cellular morphological changes”. Gross anatomy usually refers to analysis without a microscope, but the only bowel images are from tissue sections after staining and microscopic imaging.

Figure 6A: Text says, “at least 21d post-Tam”. Since muscle changes could be progressive, more precision would be desirable. What range of times after tamoxifen was evaluated? What age were mice when tamoxifen was given.

Figure 6A Legend: Text says, “(A) H&E staining (right) was performed on distal small bowel segments of Piezo1 Δ SMC and Piezo1WT mice at least 21d post-Tam, on muscularis thickness differences.” Please revise for clarity. I have no idea what this means, although I can guess from the actual figure. Also, how many mice were evaluated and how many muscle layer thickness measurements were made? What ages were the mice and what range of ages post-tamoxifen was evaluated?

Figure 6B: Authors measure SMC per unit area. Do they mean “area of muscle layers”? Were circular and longitudinal muscle layers analyzed separately or together? Were these measurements made in cross sections (as in Figure 6A) or using whole mount preparations?

Figure 6 legend: The figure includes variable numbers of asterisks to indicate level of statistical significance, but the figure legend does not indicate what these asterisks mean. Please clarify.

Figure 8A legend: Names for mRNA should be in italics. Also, text says, “qPCR calculated using $\Delta\Delta$ CT method relative to Gapdh expression.” This is confusing since “qPCR” was NOT calculated. Instead, they calculated “normalized mRNA fold expression” based on qRT-PCR results.

Figure 8A: It would be ideal if data showed absolute cycle numbers for the various channels relative to Gapdh for the WT mice instead of normalizing all of these to 1. Then, we would know how the abundance of mRNA encoding these channels compare to each other. For example, the way the data are presented, one cannot tell relative abundance of Ano1 mRNA versus Orai1 mRNA because of the WT normalization to 1, even though authors have these data. Please consider showing delta delta CT relative to Gapdh mRNA for all values or representing in another more informative manner.

Figure 8B: Is there no antibody for Orai3? All the other proteins whose mRNA are analyzed in Figure 8A are imaged at the protein level in 8B, so the omission seems odd. Also, consider arranging the images in Figure 8B to match the order of mRNA (left to right) in 8A. This would make it easier to follow. Finally, these are lovely STED images. Consider including enlargements of some regions similar to the approach in Figure 9.

Figure 8C legend: Text says, “Imaging of the muscularis from WM prepared samples stained for Ano1 and c-Kit indicates an increased Ano1 membrane-GFP marking SMCs in Piezo1 Δ SMC;mTmG mice.” This is confusing. What is “Ano1 membrane-GFP”?

Figure 9 legend says, “Distal bowel”. Do authors mean “distal small bowel”? Distal bowel might mean distal colon.

Figure 9A insert: I see a 5 micron label, but do not see a scale bar. Please review.

Figure 9B legend says, “(B) Besides the anti-IP3R (BD Bioscience), whose pseudo-yellow signal, the anti-Piezo1 antibody and other pseudo colors were similar to RyR.” I have no idea what this means. Do authors mean “Piezo1 antibody staining colocalizes with intracellular structures that also stain with antibodies to IP3R and RyR” or was there a different meaning? The next sentence says, “Piezo1 Δ SMC mice have impaired response to carbachol at steady-state in Piezo1 Δ SMC, compared to wildtype”. This is also confusing or perhaps redundant.

SFigure 1 line 976: Text says, “diameter measurements throughout the entire length of the small bowel divided into 10 portions of obstructed and unobstructed 6- to 8-week-old Piezo1WT and Piezo1 Δ SMC mice at 10d post-Tam.” I cannot find other references to bowel obstruction experiments and there is no key to indicate which line is from obstructed bowel. Where

was the obstruction? How long after obstruction were measurements made. What does the X-axis label mean relative to the position of the obstruction.

SFigure 2: What do the asterisks mean?

SFigure 3 legend seems redundant: Text says, "SMCs (DAPI, pseudowhite), interstitial cells (c-Kit), Pdgfr α + cells, glial (Gfap), and neuronal cells (Tubb3) were imaged with Leica Confocal SP8-STED microscope and stained and quantified as described in the method section. WM longitudinal samples of distal small bowel of Piezo1WT (left) and Piezo1 Δ SMC (right) mice at 21d post-Tam treatment stained with anti-c-Kit (ICC), Pdgfr α (Pdgfr α +), Gfap (Glial cells), and Tubb3 (neuronal cells, bundles, and fibers) antibodies."

SFigure 3 legend: Text says, "Tubb3 staining includes arrows to differentiate neuronal bundles (green), fibers (gray), and cells (yellow)". I do not see any gray arrows.

SFigure 4A: Lower panel scale bar is not visible.

Proposed schematic: Some of the ion channels appear to go only part way through the plasma membrane. Cav1.2 in the schematic is upside down in the figure legend compared to the way it appears in the plasma membrane.

Version 1:

Reviewer comments:

Reviewer #1

(Remarks to the Author)

Since this is a revision, I did not summarize again the major conclusions of this manuscript, but there are many novel, surprising and interesting observations. The images are remarkable and there is a lot of quantitative data, with interconnected measurements of anatomy, physiology and gene expression. This is a very valuable contribution to the literature.

The authors very carefully responded to all my prior comments and did a great job with revisions!

Unfortunately, there remain a number of very minor issues that I either missed in my prior review or that were generated during the last round of revisions. All of these require only minor changes to the text.

Minor issues:

Line 206: Says that at steady state there was reduced area under the curve in cKO (conditional Piezo1 knockout) (Figure 2C) but the figure does not show differences between WT and cKO mice at "baseline" (or at least errors bars indicating statistical differences are missing). Are "Baseline" (used in Figures 1B-I) and "Steady State" (used in Figure 1A) the same?

Line 207: Consider defining "CoV". Is this "coefficient of variation"? Since "CoV" is not used again, consider just writing it out.

Line 229: Consider "TTX, a sodium channel inhibitor that prevents action potentials in neurons". Current text suggests "TTX . . . (is) important for action potential generation".

Line 231: Says "the addition of TTX or L-NNA/ODQ did not impact the length-tension relationship in Piezo1 Δ SMC mice (Fig.3B/C)". Is this true? While the shape of the curves is similar in Figure 3A (no added factor) and Figure 3C (added L-NNA/ODQ), the Y-axis numbers are very different suggesting much less passive tension is generated in both WT and cKO mice in the presence of L-NNA/ODQ. I remain confused about why blocking NO would reduce tension, but "the data are the data".

Figures 3B, 3C, 3E, 3F: Labels indicating TTX and LNNA/ODQ like those in Figures 3H and 3I would be a nice addition, although this is clearly indicated in Figure legends, so optional.

Figures 3H, 3I, 3K and 3L are not mentioned in the Results section (I think). Figures are not mentioned in order here so perhaps I just missed the references to these figures.

Figure 4I is not mentioned in Results. All figures should be referenced in the Results.

Figure 5A: Consider changing label from "+Yoda" to "+Yoda1".

Line 339: Consider using the word "density" so text reads "proportional increase in the density of the non-SMC cellular populations" since data are cells/mm² and may not reflect an increase in absolute cell number. For example, ICC number may be unchanged, but cells/mm² increases because ICC are packed more closely together as SMC number declines. Lines 417-418 in the Discussion also suggest an increase in enteric neuron, ICC and PDGFR α + cell number, but data only show an increase in cell density.

Lines 377-388 correctly indicate, “carbachol rapidly increased active tension in Piezo1WT mice without affecting contraction duration or period (Fig.9C-E).” In contrast lines 477-478 indicate erroneously that there is “reduced responsiveness to carbachol observed in the adult small bowel (Fig.9D-F)”. I think lines 477-478 should refer to Fig. 9C (which shows reduced amplitude of carbachol response in cKO) instead of Fig. 9D-F. Also note, there is no Figure 9F.

Supplemental Figure 7: At least one image should have a scale bar.

Supplemental Figure 8: The Y-axes are labeled “RyR+ cells/ m²” and “IP3R+ cells/ m²” (using “Cell Counts” for B and D instead of “cells”). I think authors are really quantifying endoplasmic reticulum staining by antibodies or some other parameter since it seems unlikely that there are ~50 RYR+ cells in a square micron. Images clearly show (e.g., Figure 8B) that single smooth muscle cells are much larger than a single square micron.

I could not find any reference to Supplemental Figures 7 or 8 in the text. Please ensure that all figures are referenced in the Results so that readers will look at these amazing figures you spent so much time generating!

Line 613: “Between” instead of “betwee”.

Figure legend 1: Text says “Efficient Tam-induced Cre expression”. This is inaccurate. For Cre-ER, tamoxifen displaces HSP90 resulting in nuclear translocation of Cre-ER so that Cre can recombine DNA. This is described here: Lepper C, Fan CM. Generating tamoxifen-inducible Cre alleles to investigate myogenesis in mice. *Methods Mol Biol.* 2012;798:297-308. doi: 10.1007/978-1-61779-343-1_17. PMID: 22130844; PMCID: PMC3695624.

Figure legend 1D says there are 10 μm scale bars, but labels on the figure say that they are 20 μm scale bars. Please correct the figure or the legend for consistency.

Figure legend 3, lines 1013-1014: I think that they mean the “difference between WT and the Piezo SMC was smaller in the presence of TTX or L-NNA/ODQ” whereas current text reads “Piezo1ΔSMC bowel segments had (D) a greater reduction from max (peak) force at baseline that diminished in the presence of (E) TTX and (F) L-NNA/ODQ.” It appears, for example that L-NNA/ODQ had a large effect on WT (Fig 3D versus 3F) and a much smaller effect on the cKO. Please review and consider revisions if appropriate.

Figure 4 legend is confusing. For example (lines 1054-55) legend 4A says “with a dashed line to mark the operation range of 30 minutes for a 20% area increase.” Was this meant to refer to Figure 4B? Then, Figure 4B legend says “(B) IMCs isolated from external muscularis strips from 8 to 10-day-old murine pups (n=5-8 per biological sample) seeded on plastic or thermosensitive (TS) stretch-inducible hydrogels with spontaneous contractile behavior. Representative tracings depicting contractions (black) as measured by displacement (left axis) overlapped Ca²⁺ flux (red dashed) measured by absolute intensity changes, ΔF/F₀ (right axis) with (C) frequency measurements of contractions and Ca²⁺ flux shown (n>12-15 time points per group).” This description seems appropriate for Figure 4D. Please review all of the figure legends to make sure that they match the revised figures. I think 4A, 4B and 4C are all characterizing the hydrogel, and Figure 4D-F provide data about cells. For example, Figure 4D legend appears to refer to Figure 4F images. Figure 4E legend appear to refer to Figure 4G image. Then Figure 4F and 4G legends again appear to describe Figure 4F and 4G images.

Figure 8 legend title refers to Ca²⁺ ion channels, but Ano1 is a chloride channel as noted in the figure and Trpc4 is a non-specific cation channel. Recommend omitting “Ca²⁺” from the figure legend title.

Figure 8C scale bar formatting differs from the other figures and the white scale bar appears to be missing.

Figure 3 legend: How many days post-tamoxifen? What age were the mice?

POINT-BY-POINT ANSWERS TO REVIEWER COMMENTS

Reviewer #1

1. This remains an interesting topic and worthy of investigation.

A) Thank you for recognizing the importance of this topic.

2. Major changes were made to the MS, and thus it should probably be considered a new submission. The revised paper was therefore reviewed in this light, and many new questions arose beyond what had been responded to in my original review.

A) We genuinely appreciate the reviewer's acknowledgment of the extensive work performed since our previous version. Their specific suggestions drove many of the additional experiments performed, so we thank them for helping us uncover additional details of this model, currently serving as the basis of the first author's submitted Mentored Clinical Scientist Development Award (K08). While we agree that this paper has been revised significantly, we felt that several new findings, particularly the specific assessments of the muscle mechanics and the placement of Piezo1 in smooth muscle cells (SMCs) within the sarcoplasmic reticulum (SR), have only strengthened our original conclusion that loss of Piezo1 in SMCs results in small intestinal dysmotility. However, we also recognize that scientific advancements such as these spark new questions and inquiries, but to include this new data, we had to remove our prior epithelial and obstruction data, and unfortunately, we have reached the text and figure limits of the journal with our new findings that are purely focused on small bowel SMCs.

This has now been shown by:

- I. The mouse model's phenotype is delayed whole gut transit, and ex vivo wire myography assessments highlight the irregularity of the force transduced and impaired mechanics.
 - II. Intact tissue using various resolutions of confocal microscopy available to us, highlighting the changes occurring at the level of the muscularis and within the specific layers, including the abundance of related cell types (ICC, ENs, Pdgfr α +). In this latest revision, we also show that the loss of Piezo1 expression in SMCs is associated with a decline in the SMC abundance in the small bowel muscularis.
 - III. The cellular and subcellular changes in the transcription and translation of other related ion channels present at the plasma membrane and the SR. Specifically, we demonstrate the colocalization of Piezo1 with RyR and IP3R, both well-established channels in the SR with described downstream mechanisms that lead to changes in the membrane potential that would subsequently cause changes in the electrical coupling within the SIP syncytium.
 - IV. Within an "intact" network of cultured intestinal muscularis cells that we had previously published allows us to study the impact of Piezo1 deletion in SMCs within the SIP syncytium. This approach gave us a unique insight into the cell-to-cell dynamics that may be driving the tissue and phenotypic changes not achievable in isolated SMCs.
3. An **important finding** might be that Piezo1 is expressed in SR and in close association with IP3R and RYR. The super-resolution microscopy in Figs 7 and 9 are **excellent contributions** to this story. I'm not aware of any studies showing Piezo1 expression in SR membranes. So, this is a very novel aspect of the paper. The overlap of Piezo1 labeling with RYR and IP3R is fairly convincing for WT SMCs. However, another structure is also labeled and highlighted in some of these images. Generally, the RYR and IP3R labeling is punctate, suggesting very close association between Piezo1

and the SR Ca²⁺ channels. However, the larger complexes highlighted in some CM cells in Fig. 9 suggest a different organelle might also be labeled. Could these be centrosomes – the microtubule organizing centers in cells?

I am still in a quandary about what changes in tissue phenotype are directly due to alterations in SMC Piezo1 and what changes are a consequence of altered motility. I don't think the paper really drills down on this question, and it was a major concern in my last review.

A) We are grateful for the reviewers' acknowledgment of the significance of this research, as only a limited number of studies have investigated the cellular location of Piezo1 in intact tissue samples. The suggestion that Piezo1 may also be acting in the centrosomes is intriguing. Mainly since there has been some recent work highlighting the localization of Piezo1 in the centrosomes and its involvement in the mitosis of cell lines (PMID: 36574677). However, this has not yet been delineated in SMCs but is worth considering further, given that the mechanical forces critical to maintaining the integrity of centrosomes may contribute to the mechanical force sensed by Piezo1 to trigger the downstream signaling that our data is alluding to. However, we have recently found several rare halo-structures are more likely artifacts occasionally seen in intensely bright particles.

B) Piezo1 localization in other membrane-bound organelles should be considered, including the nucleus, Golgi apparatus, endosome, lysosome, mitochondria, and peroxisome. Indeed, a close examination of Figures 7 and 9 suggests that the nuclear envelope may also be a site of Piezo1 localization. However, given the strong evidence of Piezo1 co-localization with RYR and IP3R and the abundance of Ca⁺⁺ stores in the ER/SR, we decided to focus on these findings in SMCs in further detail.

This line of research aligns with the primary author's early investigative career as a Physician-Scientist (K08) in her upcoming grant application at a new institution where she hopes to obtain funding to examine additional mechanistic questions.

4. Piezo1 is expressed in SR and in close association with IP3R and RYR. This was reported, but then nothing was done about the function and significance of possible Piezo1 activation on SR Ca²⁺ release. Since this is the most novel observation, in my opinion, it is disappointing that the topic was not explored. I had many other questions about the presentation and interpretation of the study.

A) We did include an isometric force assessment that the loss of Piezo1 in SMCs has on IP3R-mediated amplitude and period consistent with the decline in IP3R expression following Piezo1 depletion (Fig. 9C-E).

5. Immuno images in Figure 1D are not very helpful. Please label the layers of the tissue and specify important points with arrows or other. β -actin is typically a housekeeping gene/protein and thus expressed in most eukaryotic cells. From these images, it seems that Piezo1 is knocked down in all cells within the 2 parallel dotted lines, thus not only from SMCs.

A) Thank you for your comment. However, we did include two dashed parallel lines indicating the location of the muscularis and the epithelial layer. We should have labeled the figure ("M" for "muscularis" and "E" for epithelium"), and have done so in response to this request.

B) The reviewer is correct that at this low magnification, it appears that Piezo1 is knocked down in all cells of the muscularis, but as subsequent (i.e., Figures 6, 7 & 9) higher magnification images reveal, the SMCs are the prominent cell type in the muscularis and also has the most abundant Piezo1 expression.

6. A positive control is needed for the statement on ln 184-185 that Piezo2 was undetectable in Piezo1 WT or KO mice.

A) Thank you for this comment. Our original statement specified that “In whole mount (WM) samples, anti-Piezo2 staining in the SMCs was undetectable in Piezo1^{WT;mTmG} and Piezo1^{ΔSMC;mTmG} mice (sFig.1C).” However, a careful review at higher resolution, it is apparent that non-SMC cells within the muscularis are positive for Piezo2 staining. The prominent Piezo2 staining comes from enteric neurons.

7. Results beginning on line 188 still cause doubts about most of the subsequent findings being directly linked to the knockdown of Piezo1. Together with Fig. 1D, it appears that the loss of Piezo1 might be broader than just SMCs, and the developmental changes (or atrophy of the SM layers) suggests tissue remodeling. This is further supported by changes in cells other than SMCs, such as increases in relative levels of ICC, Pdgfra+ cells, glia. The latter could be just a result of an increase in these cells relative to reduced numbers of SMCs.

A) The mTmG reporter line indicates that Myh11-ERT2-Cre activation is limited to SMCs and not expressed in other cell types in the muscularis, including the enteric neurons, ICCs or Pdgfra, and other cell types (see Fig.7C, and additional images in supplementary data). In **Figure 1** of this rebuttal, we further illustrate the specificity of the Myh11-ERT2-Cre activation demonstrated by the mTmG reporter. This image shows that white-colored cells (mTomato) in the muscularis are not Cre-activated and resemble Pdgfra+ cells. We have a plethora of images demonstrating that other cells in the muscularis, including ICC, glial cells, and enteric neurons, are not activated by Myh11-ERT2-Cre. This demonstrates that the Myh11-ERT2-Cre model is selective to SMCs, as we and many other investigators have described previously (PMID: 28845554). Based on the longstanding use of the Myh11-ERT2-Cre model and our imaging studies, we disagree with the reviewer's suggestion that the induced knockout is not specific to the SMCs.

Figure 1: Myh11; Piezo1-WT; mTmG mice treated with TAM, stained with anti-Piezo1 antibody (red). The membrane tomato pseudo color is white. The white-colored cells are Pdgfra cells.

B) Regarding the possibility of remodeling, it may be argued that the Cre/lox and reporter system provides the ability to discern direct vs. indirect (i.e., remodeling) changes following the induced depletion of the Lox'ed gene. Nevertheless, it is indisputable that the initiating event that sets forth these effects is the depletion of Piezo1 in SMCs, and our findings indicate that the SMCs are directly affected by the loss of Piezo1 in the SMCs on the distal small intestine.

C) Yes, we agree with the reviewer; as we suggested in line 399 of the manuscript, the uniformity of the increase in the concentration of ICC, Pdgfra+, and glia cells is likely secondary to the loss of SMCs within the muscular layer that is thinner in the Piezo1^{ΔSMC} mice. We have now confirmed that the muscularis of Piezo1^{ΔSMC} mice have a relative depletion of SMCs, and this likely accounts for the uniform increase in the ICC, Pdgfra+, and glia cells seen in these mice.

8. The large increase in ANO1 (an ICC-specific marker) suggests active remodeling of this population.

A) We agree that Ano1 in the muscularis is a fairly ICC-specific marker in wild-type mice, as seen in the SMC Transcription Browser (PMID: 30674925); however, the increase in expression of Ano1 [Fig.8A] seen in the Piezo1^{ΔSMC} mice results from an increase in SMC expression as shown in [Fig.3B/C and sFig.5]. Therefore, these changes are not attributed to a secondary remodeling event because they occur in the SMCs (mG+), not the non-Cre expressing (mT+) cells.

B) Furthermore, the behavior of ion channels located intracellularly and on the plasma membrane illustrate their truly dynamic relationship/interactions, with changes occurring immediately in response to resulting calcium dysregulation. Therefore, while Ano1 in the intestinal muscularis is primarily an ICC-specific channel, there are small but significant expression levels in SMCs (ICC 700 vs. SMC 5

FPKM) that may be non-functional in wild-type mice. However, our findings suggest that *Ano1* becomes activated in the setting of the calcium dysregulation induced by the loss of *Piezo1* in SMCs.

9. So, one cannot help but think back to the previous version of this paper where changes to many cells occur in response to *Piezo1* knockdown from SMCs, and the prior question remains: Are the many tissue level changes a direct result of *Piezo1* knockdown in SMCs or are they a response to the altered motility? Are the motility effects a result of SMC (and SM tissue) remodeling or due to changes in the other regulatory cells involved in motility? These questions are still unresolved in this new version of the paper.

A) To clarify, we included most of the transcriptional changes of the muscularis shown in Fig.8A; however, to address the reviewer's prior concerns, we performed immunofluorescent to address the cell type in the muscularis that accounts for these changes [Fig.8C]. Specifically, we did co-staining of *Ano1* with the ICC marker, c-Kit, and found that *Ano1* is primarily expressed in ICC cells in *Piezo1*^{WT} muscularis, but in *Piezo1*^{ΔSMC} mice, we see staining for the GFP+ SMCs, indicating that some of the enhanced mRNA expression in this line is related to increased expression in SMCs.

B) Regarding whether all the changes we've documented in this model are associated with the depletion of *Piezo1* in SMCs or a response to the altered motility, we believe that our analysis goes beyond the analysis that was performed in other knock-out models that result in dysmotility (i.e., *Smn*, *Trpc4*, *Trpc6*, *Cav1.2*, *Scn11a*, *Kir6.1/Sur2*, *Traak*, and others; PMID: 30972202, 19549525, 16636102, 33170808, 31542828). Given the limited analysis performed in the prior models, none included extensive staining or transcript analysis looking at these other cell types in the epithelium and muscularis; we can't draw upon the literature to help us conclude if the changes that we have documented are related to the impaired motility seen in our model.

10. I still think the title of the MS is too specific based on the data reported.

A) Yes, we agree, and we have revised the title of the manuscript.

11. The workup of contractile data in Fig 3 is a **nice addition**, but is there any likelihood that the thinning of the muscularis would not result in reduced contractility? In Figure 3, they did take steps to eliminate contributions from enteric neurons that also might have stretch-dependent mechanisms.

A) Thank you for the reviewer's compliment regarding the additional experiments performed to complete Figure 3. We attempted to address the possibility of muscularis thinning contributing to reduced contractility. A recent study in skeletal muscle highlighted the muscle atrophy that developed secondary to an interruption in *Piezo1* activity, consistent with the previous studies establishing the role of *Piezo1* over-activation in driving hypertrophy in several cell types [PMID: 35290243]. Thus, the resulting thinning of the muscularis in the setting of *Piezo1* deletion would be in line with the evolving literature. Furthermore, as suggested by reviewer #3, we have assessed and quantified the SMCs within the muscularis and found a significant depletion of SMCs in the *Piezo1*^{ΔSMC} mice (Figure 6B and sFigure 3).

12. Ln 373. *Ano1* is not a SMC Ca²⁺ channel; it is a Ca²⁺-activated Cl⁻ channel expressed specifically by ICC in GI muscles.

A) Thank you for highlighting this oversight; we had meant to indicate that these were related ion channels, not just calcium channels; this was corrected.

13. *Trpc4* and *Orai 1&3* are also not specific to SMCs. Further, *Trpc4* is generally considered a non-selective cation channel with only modest selectivity for Ca²⁺ vs monovalent cations.

A) We agree with the reviewer that *Trpc4* and *Orai 1&3* are not specific SMC markers in the intestine. Still, we have demonstrated that the loss of *Piezo1* in the SMCs results in changes in the expression of these transcripts in the muscularis, specifically in the SMCs, as determined by immunofluorescence.

14. It should also be noted that the changes noted are based on the resolution of gene or protein expression, and no functional dissection of the contributions of these channels was provided.

A) We agree with the reviewer's comment that these findings were limited to gene expression and protein localization and were not fully assessed functionally. However, this new data is intended to provide additional evidence of *Piezo1* localization in the intestinal SMCs, as requested by the reviewer's prior recommendation. This new data only further supports our original conclusions and does not contradict any previous statements. Thus, the request to investigate this further would only dilute the already very significant findings of this phenotype that was supposed to be the goal of this initial manuscript.

15. Ln 412. This does not make sense. Nothing was done to assess effects of *Piezo1* KO on enteric nerve activity. Please carefully rewrite section between Lns 410 -414 to improve clarity of the message.

A) We apologize for the confusing sentence, and the reviewer is correct in stating that we did not assess whether the *Piezo1* KO affected enteric nerve activity. We did test whether nerve activity (using TTX and L-NNA) affected the quick stretch responsiveness of the *Piezo1* KO bowel. The sentence will be written more clearly.

16. Discussion: Ln 415-420. Conclusion stated here is that ANO1 expression was turned on in SMCs by knocking-down *Piezo1*. This would be novel and surprising, as this is, as the authors point out, expressed normally by ICC. But I do not think this point has been made conclusively. My worry here is that immuno images were taken on tissues and images appear not to be exclusively single optical sections. This can cause misinterpretation of label-overlap as labeling of proteins in the Z axis (i.e., in cells above or below a specific X-Y point) can appear co-labeled. As ICC are intermixed with SMC fibers, they easily could be above or below Regions of Interest (ROI), even if an ROI is picked out that would represent a SMC (i.e. between membrane labeled with mGFP). Proper tests of this conclusion should include: imaging of single optical sections; single cell expression analysis (or expression analysis of sorted SMCs); patch clamp analysis of expression of a Ca^{2+} -activated Cl⁻ conductance in SMCs. It is a pity that whole mount ANO1 labeling, as shown for other cellular markers in supplementary figure 2, is not provided, as then we could be far more convinced whether ANO1 labeling occurs in ICC or in other cells of distinct morphology.

A) Thank you for highlighting this potential inherent limitation of the resolution distance in the vertical and lateral planes with the confocal microscope used for this data. However, we carefully defined our planes X-Y and Z planes and frequently performed Z-stack reconstruction to follow the staining of specific cells across various planes.

B) Please note that other antibody cell markers were not included in Figure 8 and sFigure 5, since the former images were obtained with the mTmG reporter where all cells other than SMCs would be mT⁺ as shown in the pseudo-white color. Therefore, we were confident that our analysis of *Ano1* localization was performed without a significant number of ICCs. However, to demonstrate this more clearly, we have performed co-staining with *Ano1* and *c-Kit* as outlined in response 9A of this rebuttal [Fig.8C]. More specifically, co-staining of *Ano1* with the ICC marker, *c-Kit*, identified that while *Ano1* is primarily expressed in ICC cells in *Piezo1*^{WT} muscularis, in *Piezo1*^{ΔSMC} mice, we see *Ano1* staining

for the GFP+ SMCs, indicating that some of the enhanced mRNA expression in this line is related to increased expression in SMCs.

17. Discussion: Ln 421. The authors speculate that ANO1 expression in SMCs would lead to an inward current by facilitating Cl⁻ efflux. However, no evidence is given for this assumption (or conclusion). In fact the equilibrium potential for Cl⁻ (E_{Cl}) has been measured to be about -40 mV (older studies by Claire Aiken and Allison Brading) and therefore depolarizations from slow waves may take membrane potential close to or even positive to E_{Cl} causing either no Cl⁻ efflux or influx. Certainly during Ca²⁺ action potentials that are superimposed upon slow waves in small intestine, Cl⁻ influx would occur, generating an outward current and inhibiting L-type Ca²⁺ channel activation.

A) We appreciate the reviewers' insight into the potential consequences of augmented Ano1 expression in SMCs and have incorporated these possibilities in the discussion section. We plan to test these various assumptions in a subsequent manuscript.

18. In general, the Schematic figure provided suggesting mechanisms for the altered SMC contractions and motility changes in SMC Piezo1 KO mice is speculative and not strongly supported by functional experiments or data. Strongly suggest removing final 2 panels describing function of ion channels, as this is completely speculation at present.

A) The diagram was intended to display the various changes in the expression of the different channels in response to the loss of Piezo1, with the suspected changes highlighted based on their known roles. We intended to only partially state that this was what was happening as additional studies have been in development to prove further the suggestions displayed in the figure. Nevertheless, we changed the Schematic figure to address some reviewer concerns.

19. Labeling on Figure 9C and sFigure 7 should be changed to IP3R not IP3, as the labeling is for the receptor not the ligand.

This has been corrected.

20. I could not resolve the Video or the Source file for the video.

We will correct this mistake.

REVIEWER #2

1. As for the major concern in my comment point 1, without direct patch clamp recording whether Piezo1 is functionally located on the plasma membrane, the claim that Piezo1 functions in the SR instead of plasma membrane, this contradicts to the dogma that Piezo1 functions as a mechanically activated cation channel in the plasma membrane to convert cell membrane tension to cationic influx.

A) We appreciate the reviewer's insistence that we perform patch-clamp recordings, but we have reviewed our findings with international patch-clamp experts at UCLA and UC Davis, who agree that the imaging data supports the notion that Piezo1 in SMCs are in the cytoplasmic organelles and assessing the Piezo1 signal would be a futile exercise. We have explored the possibility of performing patch-clamping in the SR, but this has rarely been successfully performed, and it would require isolation of the SR membrane through a process that would disrupt the cell. Given the mechanosensitive capabilities of Piezo1, we believe that such a procedure would impede its function and perhaps even its localization within the cell during the isolation process.

B) We agree that the most common models have placed Piezo1 in the plasma membrane, which has been the focus of the research thus far; however, there is some recent evidence that this type of mechanosensor may have a role in membrane-encompassing organelles within the cell. Specifically,

several studies have more recently suggested that it is located in other sites in the cell, including the ER, nucleus, and other sites [PMID: 33497620]. Furthermore, studies have found that the location of these mechanosensors may be dynamic and may change depending on the microenvironment of various tissue and cellular inputs.

C) ER membrane folding is modulated by the Lamin B receptor (LBR), which comprises a meshwork of intermediate filament proteins that form a mechanical scaffold associated with the adhesion receptors and the contractile cytoskeleton [PMID: 32302590]. Several other mechanisms have been described, including focal adhesion-based and myosin II-dependent mechanisms that may contribute to the stretching of the ER membrane. This membrane is continuous with the nuclear membrane that forms the nuclear envelope, suggesting the potential critical role of Piezo1 as a component of this process.

D) Others have found that IP3Rs may serve a role in stretch-activated Ca⁺⁺ release from the nucleus and ER, but this process is likely tissue and cellular-type dependent. Furthermore, our finding that IP3Rs in the ER are Piezo1-dependent is novel and underscores a potential connection of ER-based Piezo1 in stretch or compression-induced changes in SMCs [PMID: 33060332].

2. According to the authors new claim that Piezo1 might function together with IP3R and RyR in the SR, then how Piezo1 is activated? Can the authors observe Piezo1-dependent Ca²⁺ sparks released from SR?

A) We want to reiterate that the schematic was a **proposed** sequence of events based on the changes in expression levels in our data and the known functional roles of each channel. However, as alluded to above, this hypothesis and others will be tested by the first author as part of her upcoming K08 grant submission in collaboration with Fernando Santana and Rose E. Dixon. Therefore, it would not be appropriate nor necessary to include in this current manuscript, given the extent of data already included.

REVIEWER #3

1. This is a nicely written manuscript with a lot of data describing effects of Piezo1 loss in bowel smooth muscle. The data define a previously unrecognized role for Piezo1 in myogenic contractility responses of the bowel, a potential therapeutic target to treat a variety of bowel motility disorders.

In this revision, data about epithelial biology and bowel obstruction were omitted so authors could more thoroughly investigate (and present data about) how Piezo1 impacts smooth muscle biology, as suggested by the other reviewers. This seems reasonable.

The authors present lot of new data: Figure 3B, parts of Figure 6, Figure 9 and 10, Supplemental figures s1A and B, s3/4, s6, s7, and s8, a new movie, along with a new schematic (assuming I understand correctly) are all new. There are 7 new antibodies, 3 new chemical inhibitor studies, and new data about animal husbandry. I commend them on all this additional work that helps clarify the subcellular location and function of Piezo1 in bowel smooth muscle.

I do not see any major flaws in the manuscript.

Minor findings include:

- a. Their model results in loss of Piezo1 in most bowel smooth muscle cells. There is a compensatory increase in Piezo2 in small bowel but not in other regions tested.
- b. Loss of smooth muscle Piezo1 causes weight loss, slower bowel transit, reduced stool output, and reduced food consumption.
- c. Smooth muscle layers are thinner after Piezo1 loss, but density of PDGFRa+ cells, ICC, enteric glia, and enteric neurons all increase for reasons that are not clear.

A) We appreciate the reviewer's recognition that many changes were made to this latest version of the manuscript and agree that the new data provides a more in-depth examination of Piezo1's role in SMCs.

Recommendations:

2. It would be nice to know if there are fewer smooth muscle cells per mm² or if the individual smooth muscle cells are smaller. These are data that they could obtain easily from their beautiful whole mount preparations. Assuming smooth muscle cells are less abundant or smaller, does the reduction in the number or size of smooth muscle cells explain why all the other cells evaluated in Figure 6 are more densely packed together? This would provide a simple explanation and seems more likely than Piezo1 loss-induced cell division for each of these other cell types.

A) We thank the reviewer for his helpful suggestions. We performed this analysis and found that there are fewer SMCs in the muscularis of Piezo1^{ΔSMC} mice than in Piezo1^{WT} mice. Our prior examination of apoptosis based on staining with anti-cleaved caspase 3 antibody failed to identify any differences between the two groups. The reduction of SMCs in the Piezo1^{ΔSMC} muscularis may account for the general increase in the abundance of ICC, PDGFRA, Glial and enteric neurons in this group.

3. Levels of many calcium channels and the Ano1 channel change is SMC after Piezo1 loss. The reason for these changes is not clear and not investigated. I do not believe there is a simple study to do that would tell us why the levels of these other calcium channels change, and this seems like a topic that could be explored in the future, should they care to pursue this.

A) We agree with the reviewer that the reason for these changes in channel number induced by the loss of Piezo1 in the SMCs is an interesting question. It will be one of the questions that the first author, a young physician-scientist, hopes to pursue in **her submitted K08 grant application**; however, this manuscript must be published if she hopes to obtain competitive funding.

4. Loss of Piezo1 from SMC did not prevent response to carbachol or 5-HT, although the magnitude of the frequency response to carbachol might be higher in Piezo1 conditional KO than in WT (Figure 5B). Statistical comparisons between WT and KO are not made in Figure 5.

A) Thank you for this suggestion, and statistical analysis was performed between the two groups.

5. Confocal super-resolution microscopy (SP8-STED/FLIM/FCSSP8 imaging) shows Piezo1 is not in plasma membrane, but instead in intracellular compartments. The absence of Piezo1 antibody signal in tamoxifen-treated conditional mutant mice and the use of mTmG are elegant controls for antibody specificity and defines cells that have undergone Cre-mediated DNA recombination. Images convincingly show colocalization of Piezo1 with RyR and IP3R in the sarcoplasmic reticulum but do not show physical interaction between these channels as suggested in the schematic.

A) We appreciate the reviewer's recognition of the convincing confocal super-resolution microscopy imaging that demonstrates co-localization of Piezo1 with RyR and IP3R; however, we mistakenly showed direct interactions in our schematic overview. This has been changed.

6. Interestingly, IP3R protein immunohistochemical signal declines in the Piezo1-depleted SMC while RyR abundance is maintained in the absence of Piezo1. Consistent with this observation, carbachol induced contraction amplitude is also markedly reduced in Piezo1 conditional KO mice. Curiously, amplitude of carbachol-induced contractions seems to decline by ~ 4.5-fold in Figure 9C-E, but the reduction in calcium transient amplitude (Figure 5A) is not nearly as dramatic in Piezo1 conditional KO bowel. Any thoughts about this discrepancy between calcium transient amplitude and bowel contraction amplitude (no comment needed, just for authors to consider)?

A) Thank you for your comment; we believe the reviewer may be referring to Figure 5A instead of Figure 1A, which does not refer to calcium transient amplitude. These calcium transients were

obtained from cultured intestinal muscularis cells from neonatal mouse pups, while the force transduction *ex vivo* data was obtained from adult mice 21 days post-Tam; therefore, these differences may be developmentally regulated, or time-dependent. We should also emphasize that Piezo1 depletion in the *in vitro* pup model depends on the administration of the Tam derived over 24 hrs, and then the functional assessment is 5 days later. In contrast, the adult mouse model demonstrates thinning of the muscularis only after 10 days, and therefore, their likely loss or lack of replacement of SMCs following Piezo1 depletion may only be modeled in *in vivo*, and not in pup in our *in vitro* model.

7. Although they addressed all my comments in this round of revision, there are still many minor issues to address, and I am not sure I found all of them. Careful re-reading is encouraged.

A) We appreciate the reviewers thoughtful suggestions that have helped us further improve each subsequent version of our submitted manuscript. We will make these changes and also utilize an objective reviewer to ensure no additional mistakes are made that take away from the significance of the manuscript.

DIVISION OF GASTROENTEROLOGY
Department of Pediatrics
David Geffen School of Medicine at UCLA
10833 Le Conte Ave, Box 951752
Los Angeles, California 90095-1752
Tel: (310) 794-5532

Re: COMMSBIO-24-5531-T

December 25, 2024

Point-by-Point Response: Reviewer #1 (Remarks to the Author): Response to Reviewers Comments

Specific comments, with recommendations to address each comment. There are many minor issues to address:

1. Line 112: "modulates" instead of "modulating". This has been corrected.
2. Line 115: "SIPgenic innervation" is an odd term that I cannot easily find referenced. I say "odd" because innervation, by definition, is something nerves do, and none of the SIP syncytium cells are nerves. I guess this is OK, but it might be more confusing than helpful for readers who think of "innervation" the way that I do. This was a term adapted from the recently coined term that focuses on the uniqueness of the SIP syncytium, we have modified it to remove the word innervation to prevent confusion.
3. Line 175: Says Piezo2 mRNA is reduced in bladder, but the graph shows Piezo 2 mRNA is slightly increase in the Piezo cKO mouse. I am confused. Apologies for the oversight, you are correct Piezo2 is mildly increased in the Piezo1 cKO mouse, this has been corrected and clarified in the manuscript.
4. Line 206: Text says, "Piezo1 modulates the strength and pattern of contractions in response to mechanical forces, likely by contributing to the Cav1.2-mediated Ca²⁺ influx that triggers contractility." This is confusing. Data show loss of Piezo1 alters strength and patterns of contractions, but data in the preceding paragraph do not show calcium transients. Also, how might Piezo1 "contribute to Cav1.2-mediated Ca²⁺ influx"? Unless authors wish to add supporting data, I recommend omitting this speculation from the Results section. Thank you for pointing out. We have removed the last statement regarding the contribution to Cav1.2 out of the results section to improve clarity.
5. Lines 212-214: Text says, "The addition of tetrodotoxin (TTX), an inhibitor of sodium channels essential for the generation of action potentials in neurons, and L-NNA, an antagonist of nNOS combined with ODQ, a soluble guanylyl cyclase (sGC) inhibitor to isolate EN modulation, did not impact the length-tension relationship in Piezo1ΔSMC mice (Fig.3B/C)." This does not appear to be true. While the graphs in Figures 3A-C appear similar, the scale bars differ. I am also confused about why blocking neural signaling with TTX or blocking NO signaling in Fig 3C should reduce passive tension. Since NO is a major smooth muscle relaxant, I expected blocking NO (or blocking all nerve activity) to increase muscle tension. Is there a simple way for readers to understand these data? We performed these additional studies based on the suggestions of the other reviewers to isolate true myogenic contributions separate from possible neuronal impact. We tried to phrase it similarly to how it has previously been written but we have rephrased it further to improve clarity.
6. Lines 251-252: Text says, "Overall, these data show that in an isolated in vitro model of IMCs, loss of Piezo1 in SMCs results in a disruption in Ca²⁺ flux, likely related to disruptions in the electrical coupling within the SIP syncytium". While they show loss of Piezo1 in SMC impacts calcium flux, I do not see evident of "disruption in electrical coupling within the SIP syncytium", which would presumably mean disruption of gap junctions, a topic for which there are no data. Please rephrase to avoid speculating far beyond the data, or provide data that electrical coupling is disrupted in the SIP syncytium after Piezo1 loss. Thank you, this has been modified in the text.

7. Lines 259-260: Text says, "We used the mTmG reporter mice to assess whether SMC depletion of Piezo1 SMCs in the muscularis following Tam." This seems like an incomplete thought. Please review. Thanks for the suggestion, and we have modified the text.
8. Figure 8C: It would help to see the individual colors as separate panels, as authors did for other figures. These are available in supplementary figures 5 and 6.
9. Line 304: Text says, "there was a significant reduction in IP3R expression in Piezo1 Δ SMC;mTmG compared to that in Piezo1WT;mTmG SMCs." Where are the data about IP3R abundance? To support this statement, I expected a figure to show qPCR data or quantitative immunohistochemistry. This seems like an important omission since authors again comment on reduced IP3R expression on lines 322, 376 and 381. Thanks for the suggestion to quantify the differences in IP3R and RyR abundance in KO and WT mice. We have completed this task using AIVIA software, and is now incorporated in Figure 9c.
10. Lines 359-361: Text says, "While current models have predominantly placed Piezo1 primarily in the plasma membrane, supports its presence within subcellular organelles in other mechanically active organs." This sentence seems to be missing one or more words. Thank you, this has been modified in the text.
11. Line 376: Text says, "Our novel finding that IP3Rs in the ER are Piezo1-dependent". What are the data to support this statement? We have corrected this statement by suggesting a correlation and not an explicit dependency of IP3R for Piezo1 in SMCs.
12. Line 490: Text says, "Protein was extracted from tissue and lysed". What does it mean for protein to be "lysed"? I suspect authors mean "Protein was extracted from tissue and boiled in Laemmli buffer". Please clarify. This is correct, and it was corrected.
13. Line 502: Text says, "Paraffin-embedded tissues were deparaffinized on a heat block at 50°C for 10 min, then quickly transferred into a container containing xylene". The phrase "container containing" seems redundant. Also, it would be more accurate to say "deparaffinize with xylene" since heating to 50C does not remove paraffin. Thank you. This has been modified in the text.
14. Lines 505-506: Text says, "quenched in 0.3% H2O2". Would it be more accurate to say, "oxidized with" or "treated with", since it is not clear what hydrogen peroxide would "quench" in this context. Thank you; this was a helpful suggestion. We modified the text and used the term "treated with."
15. Line 529: Consider changing "interstitial" to "interstitial cells of Cajal", a specific cell type. Thank you. This has been modified in the text.
16. Lines 845-849: Text says, "A) Myh11-Cre expression confirmed with Leica demonstrating distal small bowel segment from Piezo1WT mice containing the membrane-associated Tdtomato fluorescence (mTmG) reporter (Piezo1WT;mTmG) revealed Tdtomato fluorescence in all cell types across the layers of the bowel, and no membrane-GFP fluorescence (pseudo-green), indicating an absence of Cre expression without tamoxifen induction." [This is confusing. What does "confirmed with Leica" mean? What are each of the 4 panels intended to show? Consider adding labels across the top to indicate what color was imaged. How is "Myh11-Cre expression confirmed" in panel (A) in the absence of added tamoxifen? Mouse line name is also inaccurate since this is "Myh11-ERT2/Cre". How old are these mice?]

Thank you. We have cleaned up the wording in this section and added labels for each panel. In the first sentence of the Results section, we clarified the name and abbreviation of our various mouse lines.

- wildtype (WT), *Piezo1*^{WT/WT} - *Myh11*^{ERT2/Cre} (**Piezo1**^{WT}) and homozygote knockout (KO), *Piezo1*^{f/f} - *Myh11*^{ERT2/Cre} (**Piezo1** ^{Δ SMC})
- To specify cells with Cre induction, mice were also generated with mTmG reporter expressing membrane-targeted tdTomato+ (mT) or EGFP (mG) (**Piezo1**^{WT;mTmG} and **Piezo1** ^{Δ SMC;mTmG})

17. Lines 849 to 852: Text says, "B) In contrast, 21d post-Tam treated Piezo1WT;mTmG mice, the epithelial layer retained Tdtomato, while the cellular components of the muscularis were either Tdtomato or GFP+." Were these

mice the same age as the mice in panel (A)? Thank you, we improved the legends so that the age and timing are clearer, and indicated that the mice in the B section are similar to those in section A.

18. Figure 1C: What age were the mice? How long after tamoxifen were analyses done? Thank you, this has been clarified in the figure legends.
19. Figure 1F: Was Carmine dye given “orally” as stated (which usually means “by mouth”) or was Carmine given by gavage (into the stomach). Changed to the term “oral gavage” which is what is the language we have in our IACUC.
20. Figure 2A in the green box: The “i” (amplitude) is not labeled on the figure. A “v” is labeled on the figure but the meaning of this “v” is not described in the figure legend. Also, black and green tracings are difficult to distinguish. Please consider adding labels or using colors that are more easily distinguished, including by color blind individuals. Parts of Figure 2A outside the green box are not described. Thanks for this suggestion, we have made these changes.
21. Figure 2 Legend: Text says, “Phasic contractile activity, including (B) amplitude, (C) AUC, (D) duration, and (E) period, were altered in Piezo1 Δ SMC mice compared to controls (n=50 samples, N=3 mice per group).” This is confusing. Did authors measure >15 ileal rings (3mm long each) per mouse? How is this possible? Alternatively, were 3 ileal rings evaluated per genotype after stretching each piece > 15 times? Sorry for the confusion. For each 3mm ileal ring, we can stretch it up to 5 times before the tissue breaks, so for several analyses we used change in force depending on the specific characteristic we were testing for. Following each step, we are able to gather 2 minute increments of data as long as our baseline remains so attempted to match these force measurements simultaneously with a knockout in the other chamber. We initially struggled with an approach to the wire myography but have based this on the approach taken by the Reno group that have published on this method previously.
22. Figure 3A-C: How is “internal circumference” measured while the tissue is in a pressure transducing apparatus? The intestine is placed on two wires that are pre-measured a specific distance and each turn of the knob results in a specific distance so we were able to calculate the change in the internal diameter or circumference.
23. Figure 3G-L: Y axis labels say “%Forec/s”. Should this say “%Force/s”? If so, the error was made twice. Thank you, this has been modified in the figure.
24. Figure 3I and 3L: In the PDF version of the figures, labels above the graphs overlap with data points and the “ODQ (1 μ M)” is not rendering properly. Also, is there a reason for not analyzing at least 3 mice per group for Figure 3G and 3J? Thank you, this has been modified in the text to more accurately include data collected from 3 different mice at 3 different dates.
25. Figure 6 Legend title: Text says, “Gross and cellular morphological changes”. Gross anatomy usually refers to analysis without a microscope, but the only bowel images are from tissue sections after staining and microscopic imaging. This was corrected.
26. Figure 6A: Text says, “at least 21d post-Tam”. Since muscle changes could be progressive, more precision would be desirable. What range of times after tamoxifen was evaluated? What age were mice when tamoxifen was given. Mice between the ages of 4–8 weeks old were given Tam, and euthanized 21 to 28 days following the last dose of Tam. This was stated more clearly in the text.
27. Figure 6A Legend: Text says, “A) H&E staining (right) was performed on distal small bowel segments of Piezo1 Δ SMC and Piezo1WT mice at least 21d post-Tam, on muscularis thickness differences.” Please revise for clarity. I have no idea what this means, although I can guess from the actual figure. Also, how many mice were evaluated and how many muscle layer thickness measurements were made? What ages were the mice and what range of ages post-tamoxifen was evaluated? Thank you, we had included this in detail in the methods and results but have now incorporated it into the legend for clarity.

28. Figure 6B: Authors measure SMC per unit area. Do they mean “area of muscle layers”? Were circular and longitudinal muscle layers analyzed separately or together? Were these measurements made in cross sections (as in Figure 6A) or using whole mount preparations? This was performed by analyzing WM preparations as described.
29. Figure 6 legend: The figure includes variable numbers of asterisks to indicate level of statistical significance, but the figure legend does not indicate what these asterisks mean. Please clarify. Thank you. We have now updated all the graphs so that the numerical version of the p-value is displayed. Thus, this no longer applies.
30. Figure 8A legend: Names for mRNA should be in italics. Also, text says, “qPCR calculated using $\Delta\Delta$ CT method relative to Gapdh expression.” This is confusing since “qPCR” was NOT calculated. Instead, they calculated “normalized mRNA fold expression” based on qRT-PCR results. Yes, this sentence was correct and now says: “Normalized mRNA fold expression based on qRT-PCR results was calculated using the \$\Delta\Delta\$ CT method relative to Gapdh expression.”
31. Figure 8A: It would be ideal if data showed absolute cycle numbers for the various channels relative to Gapdh for the WT mice instead of normalizing all of these to 1. Then, we would know how the abundance of mRNA encoding these channels compare to each other. For example, the way the data are presented, one cannot tell relative abundance of Ano1 mRNA versus Orai1 mRNA because of the WT normalization to 1, even though authors have these data. Please consider showing delta delta CT relative to Gapdh mRNA for all values or representing in another more informative manner. This is a helpful suggestion, and we have modified the order of the transcripts and provided the delta delta CT in Supplement 5A.
32. Figure 8B: Is there no antibody for Orai3? All the other proteins whose mRNA are analyzed in Figure 8A are imaged at the protein level in 8B, so the omission seems odd. Also, consider arranging the images in Figure 8B to match the order of mRNA (left to right) in 8A. This would make it easier to follow. Finally, these are lovely STED images. Consider including enlargements of some regions similar to the approach in Figure 9. While there are antibodies to Orai3, budgetary restrictions limited our assessment to Orai1. We have rearranged the antibody staining images to match the order of the qPCR panel.
33. Figure 8C legend: Text says, “Imaging of the muscularis from WM prepared samples stained for Ano1 and c-Kit indicates an increased Ano1 membrane-GFP marking SMCs in Piezo1 Δ SMC;mTmG mice.” This is confusing. What is “Ano1 membrane-GFP”? We have explained this more clearly: (C) Imaging of the muscularis from WM-prepared samples stained for Ano1 and c-Kit indicates an increase of Ano1 staining along the GFP+ plasma membrane of SMCs in Piezo1 ^{\$\Delta\$ SMC;mTmG} mice.
34. Figure 9A insert: I see a 5 micron label, but do not see a scale bar. Please review. This has been added.
35. Figure 9B legend says, “(B) Besides the anti-IP3R (BD Bioscience), whose pseudo-yellow signal, the anti-Piezo1 antibody and other pseudo colors were similar to RyR.” I have no idea what this means. Do authors mean “Piezo1 antibody staining colocalizes with intracellular structures that also stain with antibodies to IP3R and RyR” or was there a different meaning? The next sentence says, “Piezo1 Δ SMC mice have impaired response to carbachol at steady-state in Piezo1 Δ SMC, compared to wildtype”. This is also confusing or perhaps redundant. Thanks for pointing out these redundancies, which we have corrected.
36. Figure 1 line 976: Text says, “diameter measurements throughout the entire length of the small bowel divided into 10 portions of obstructed and unobstructed 6- to 8-week-old Piezo1WT and Piezo1 Δ SMC mice at 10d post-Tam.” I cannot find other references to bowel obstruction experiments and there is no key to indicate which line is from obstructed bowel. Where was the obstruction? How long after obstruction were measurements made. What does the X-axis label mean relative to the position of the obstruction. We have corrected this information and have removed any discussion related to obstruction. This data is now in Supplement 1 D and E.
37. SFigure 2: What do the asterisks mean? The asterisks have been removed.
38. SFigure 3 legend seems redundant: Text says, “SMCs (DAPI, pseudowhite), interstitial cells (c-Kit), Pdgfra+ cells, glial (Gfap), and neuronal cells (Tubb3) were imaged with Leica Confocal SP8-STED microscope and stained and quantified as described in the method section. WM longitudinal samples of distal small bowel of Piezo1WT (left)

and Piezo1 Δ SMC (right) mice at 21d post-Tam treatment stained with anti-c-Kit (ICC), Pdgfra (Pdgfra+), Gfap (Glial cells), and Tubb3 (neuronal cells, bundles, and fibers) antibodies.” Yes, the redundancy was corrected.

39. SFigure 3 legend: Text says, “Tubb3 staining includes arrows to differentiate neuronal bundles (green), fibers (gray), and cells (yellow)”. I do not see any gray arrows. This error was corrected.

40. SFigure 4A: Lower panel scale bar is not visible. This error was corrected.

41. Proposed schematic: Some of the ion channels appear to go only part way through the plasma membrane. Cav1.2 in the schematic is upside down in the figure legend compared to the way it appears in the plasma membrane. This was purposely done depending on the proposed location and potential function of the channels suggested by our data and based on the literature we were illustrating, depending on where the Ca²⁺ flux is being driven to. I adjusted the legend to match, but the specific location of Piezo1 within the intracellular region is yet to be determined with a more precise methodology.

Kind Regards,

Martín G. Martín M.D., M.P.P.

Professor, Department of Pediatrics, Gastroenterology
University of California, Los Angeles

Geoanna M. Bautista, M.D.

Assistant Professor, Department of Pediatrics, Neonatology
University of California, Davis

POINT-BY-POINT RESPONSE TO REVIEWERS' COMMENTS:

Reviewer #1 (Remarks to the Author):

1. Line 206: Says that at steady state there was reduced area under the curve in cKO (conditional Piezo1 knockout) (Figure 2C) but the figure does not show differences between WT and cKO mice at “baseline” (or at least errors bars indicating statistical differences are missing). Are “Baseline” (used in Figures 1B-I) and “Steady State” (used in Figure 1A) the same?
Thank you for pointing this out, the graph has been updated to include all comparisons since that one had accidentally been left out. Steady state is when contraction behavior has reached its most consistent state (detected forces are no longer drastically increasing or decreasing). All measurements in this figure are taken after the tissue has reached steady state. Baseline refers to the contraction behavior characterized at steady state (amplitude, period, etc.), prior to stretch experiments/modification experiments.
2. Line 207: Consider defining “CoV”. Is this “coefficient of variation”? Since “CoV” is not used again, consider just writing it out. This has been corrected in the manuscript.
3. Line 229: Consider “TTX, a sodium channel inhibitor that prevents action potentials in neurons”. Current text suggests “TTX . . . (is) important for action potential generation”. This has been corrected.
4. Line 231: Says “the addition of TTX or L-NNA/ODQ did not impact the length-tension relationship in Piezo1 Δ SMC mice (Fig.3B/C)”. Is this true? While the shape of the curves is similar in Figure 3A (no added factor) and Figure 3C (added L-NNA/ODQ), the Y-axis numbers are very different suggesting much less passive tension is generated in both WT and cKO mice in the presence of L-NNA/ODQ. I remain confused about why blocking NO would reduce tension, but “the data are the data”.
Unfortunately, due to inherent variations in the animals despite controlling for age, timing and the need for constant calibration of the wire myography system, we based our assessments on the change from baseline behavior rather than the raw force measurements, applying a similar approach as other studies using wire myography have previously applied. To further minimize potential confounding variables, we performed all assessments on paired wildtype and cKO specimens from age-matched mice on the same day, with the same investigator, using solutions prepared immediately prior to the experimental runs, and run simultaneously using the multi-wire myography set-up. By looking at change from baseline rather than raw force values for each tissue sample, we are able to do a better compare between different samples.
5. Figures 3B, 3C, 3E, 3F: Labels indicating TTX and LNNA/ODQ like those in Figures 3H and 3I would be a nice addition, although this is clearly indicated in Figure legends, so optional. I'm not sure if the wrong figure was transmitted, but we did add those identical titles to all 3B, 3C, 3E, 3F as you had suggested previously. We will double check to make sure the old figures have been removed completely so there's no error. Thank you!
6. Figures 3H, 3I, 3K and 3L are not mentioned in the Results section (I think). Figures are not mentioned in order here so perhaps I just missed the references to these figures. We apologize for the confusion. We struggled with this specific figure in terms of how to make it clear visually without being redundant but the reason we performed those additional assessments were for the same goal of blocking neuronal activity to isolate as much of the muscle-specific drivers that may impact our interpretation.

7. Figure 4I is not mentioned in Results. All figures should be referenced in the Results. Thank you for pointing that out, the last sentence referencing 4H should have included 4I since that was the quantification of the representative image shown in 4H.
8. Figure 5A: Consider changing label from “+Yoda” to “+Yoda1”. When we first submitted this paper, Yoda2 had not yet been well-established, but we have updated this to minimize confusion.
9. Line 339: Consider using the word “density” so text reads “proportional increase in the density of the non-SMC cellular populations” since data are cells/mm² and may not reflect an increase in absolute cell number. For example, ICC number may be unchanged, but cells/mm² increases because ICC are packed more closely together as SMC number declines. Lines 417-418 in the Discussion also suggest an increase in enteric neuron, ICC and PDGFRa+ cell number, but data only show an increase in cell density. This is an important suggestion, and we have incorporated it in the revised manuscript.
10. Lines 377-388 correctly indicate, “carbachol rapidly increased active tension in Piezo1WT mice without affecting contraction duration or period (Fig.9C-E).” In contrast lines 477-478 indicate erroneously that there is “reduced responsiveness to carbachol observed in the adult small bowel (Fig.9D-F)”. I think lines 477-478 should refer to Fig. 9C (which shows reduced amplitude of carbachol response in cKO) instead of Fig. 9D-F. Also note, there is no Figure 9F. This has been corrected in the manuscript.
11. Supplemental Figure 7: At least one image should have a scale bar. This been corrected
12. Supplemental Figure 8: The Y-axes are labeled “RyR+ cells/μm²” and “IP3R+ cells/μm²” (using “Cell Counts” for B and D instead of “cells”). I think authors are really quantifying endoplasmic reticulum staining by antibodies or some other parameter since it seems unlikely that there are ~50 RYR+ cells in a square micron. Images clearly show (e.g., Figure 8B) that single smooth muscle cells are much larger than a single square micron. This has been corrected.
13. I could not find any reference to Supplemental Figures 7 or 8 in the text. Please ensure that all figures are referenced in the Results so that readers will look at these amazing figures you spent so much time generating! Thank you for pointing that out, these were the expanded versions of the earlier figure and was referenced along with that original figure since it was to provide the single channel data, but realized we combined the figure number likely for spacing limits but went ahead and fixed that.
14. Line 613: “Between” instead of “betwee”. This has been corrected.
15. Figure legend 1: Text says “Efficient Tam-induced Cre expression”. This is inaccurate. For Cre-ER, tamoxifen displaces HSP90 resulting in nuclear translocation of Cre-ER so that Cre can recombine DNA. This is described here: Lepper C, Fan CM. Generating tamoxifen-inducible Cre alleles to investigate myogenesis in mice. *Methods Mol Biol.* 2012;798:297-308. doi: 10.1007/978-1-61779-343-1_17. PMID: 22130844; PMCID: PMC3695624. The correction was made.
16. Figure legend 1D says there are 10 μm scale bars, but labels on the figure say that they are 20 μm scale bars. Please correct the figure or the legend for consistency. The correction was made.
17. Figure legend 3, lines 1013-1014: I think that they mean the “difference between WT and the PiezoΔSMC was smaller in the presence of TTX or L-NNA/ODQ” whereas current text reads “Piezo1ΔSMC bowel segments had (D) a greater reduction from max (peak) force at baseline that diminished in the presence of (E) TTX and (F) L-NNA/ODQ.” It appears, for example that L-NNA/ODQ

had a large effect on WT (Fig 3D versus 3F) and a much smaller effect on the cKO. Please review and consider revisions if appropriate. Thank you for pointing out the confusion in this description. We have made the recommended revisions in the text.

18. Figure 4 legend is confusing. For example (lines 1054-55) legend 4A says “with a dashed line to mark the operation range of 30 minutes for a 20% area increase.” Was this meant to refer to Figure 4B? Then, Figure 4B legend says “(B) IMCs isolated from external muscularis strips from 8 to 10-day-old murine pups (n=5-8 per biological sample) seeded on plastic or thermosensitive (TS) stretch-inducible hydrogels with spontaneous contractile behavior. Representative tracings depicting contractions (black) as measured by displacement (left axis) overlapped Ca²⁺ flux (red dashed) measured by absolute intensity changes, $\Delta F/F_0$ (right axis) with (C) frequency measurements of contractions and Ca²⁺ flux shown (n>12-15 time points per group).” This description seems appropriate for Figure 4D. Please review all of the figure legends to make sure that they match the revised figures. I think 4A, 4B and 4C are all characterizing the hydrogel, and Figure 4D-F provide data about cells. For example, Figure 4D legend appears to refer to Figure 4F images. Figure 4E legend appears to refer to Figure 4G image. Then Figure 4F and 4G legends again appear to describe Figure 4F and 4G images. Thank you for pointing out this inconsistency. The labels have been shifted to the correct positions in the legend so that the following text describes the correct graph.
19. Figure 8 legend title refers to Ca²⁺ ion channels, but Ano1 is a chloride channel as noted in the figure and Trpc4 is a non-specific cation channel. Recommend omitting “Ca²⁺” from the figure legend title. This has been corrected.
20. Figure 8C scale bar formatting differs from the other figures and the white scale bar appears to be missing. This has been corrected.
21. sFigure 3 legend: How many days post-tamoxifen? What age were the mice? All assessments had been done for the same age-matched mice that were treated with Tamoxifen between 4-6 weeks and assessed 3 weeks following Tam or sham treatment, thus were assessed between 7-9 weeks. We have included this information in here that was added to our earlier figures for clarity.